# Any2Graph: Deep End-To-End Supervised Graph Prediction With An Optimal Transport Loss

**Paul Krzakala**
LTCI & CMAP , Télécom paris, IP Paris

**Junjie Yang**
LTCI, Télécom paris, IP Paris

**Rémi Flamary**
CMAP, Ecole polytechnique, IP Paris

**Florence d'Alché-Buc**
LTCI, Télécom paris, IP Paris

**Charlotte Laclau**
LTCI, Télécom paris, IP Paris

**Matthieu Labeau**
LTCI, Télécom paris, IP Paris

## Abstract

We propose Any2Graph, a generic framework for end-to-end Supervised Graph Prediction (SGP) i.e. a deep learning model that predicts an entire graph for any kind of input. The framework is built on a novel Optimal Transport loss, the Partially-Masked Fused Gromov-Wasserstein, that exhibits all necessary properties (permutation invariance, differentiability) and is designed to handle any-sized graphs. Numerical experiments showcase the versatility of the approach that outperforms existing competitors on a novel challenging synthetic dataset and a variety of real-world tasks such as map construction from satellite image (Sat2Graph) or molecule prediction from fingerprint (Fingerprint2Graph). [1]

## 1 Introduction

This work focuses on the problem of Supervised Graph Prediction (SGP), at the crossroads of Graph-based Learning and Structured Prediction. In contrast to node and graph classification or link prediction widely covered in recent literature by graph neural networks, the target variable in SGP is a graph and no particular assumption is made about the input variable. Emblematic applications of SGP include knowledge graph extraction [28] or dependency parsing [17] in natural language processing, conditional graph scene generation in computer vision [53], [12], or molecule identification in chemistry [10, 55, 49], to name but a few. Moreover, close to SGP is the unsupervised task of graph generation notably motivated by *de novo* drug design [8, 15, 41].

SGP raises some specific issues related to the complexity of the output space and the absence of widely accepted loss functions. First, the non-Euclidean nature of the output to be predicted makes both inference and learning challenging while the size of the output space is extremely large. Second, the arbitrary size of the output variable to predict requires a model with a flexible expressive power in the output space. Third, graphs are characterized by the absence of natural or ground-truth ordering of their nodes, making comparison and prediction difficult. This particular issue calls for a node permutation invariant distance to predict graphs. Scrutinizing the literature through the lens of these issues, we note that existing methodologies circumvent the difficulty of handling output graphs in various ways. A first body of work avoids end-to-end learning by relying on some relaxations. For instance, energy-based models (see for instance [38]) convert the problem into the learning of an

---

[1]All code is available at `https://github.com/KrzakalaPaul/Any2Graph`.

energy function of input and output while surrogate regression methods [10] implicitly embed output graphs into a given Hilbert space where the learning task boils down to vector-valued regression. Note that these two families of approaches generally involve a rather expensive decoding step at inference time. In what follows, we focus on methods that directly output graphs or close relaxations, enabling end-to-end learning.

One strategy to overcome the need for a permutation invariant loss is to exploit the nature of the input data to determine a node ordering, with the consequence that application to new types of data requires similar engineering. For instance, in *de novo* drug generation SMILES representations [8] are generally used to determine atom ordering. In semantic parsing, the target graph is a tree that can be serialized [3] while in text-to-knowledge-graph, the task is re-framed into a sequence-to-sequence problem, often addressed with large autoregressive models. Finally, for road map extraction from satellite images, one can leverage the spatial positions of the nodes to define a unique ordering [5].

Another line of research proposes to address this problem more directly by seeking to solve a graph-matching problem, i.e., finding the one-to-one correspondence between nodes of the graphs. Among approaches in this category, we note methods dedicated to molecule generation [25] where the invariant loss is based on a characterization of graphs, ad-hoc to the molecule application. While being fully differentiable their loss does not generalize to other applications. In the similar topic of graph generation, Simonovsky and Komodakis [37] propose a more generic definition of the similarity between graphs by considering both feature and structural matching. However, they solve the problem using a two-step approach by using first a smooth matching approximation followed by a rounding step using the Hungarian algorithm to obtain a proper one-to-one matching, which comes with a high computational cost and introduces a non-differentiable step. For graph scene generation, Relationformer [36] is based on a bipartite object matching approach solved using a Hungarian matcher [11]. The main shortcoming of this approach is that it fails to consider structural information in the matching process. The same problem is encountered by Melnyk et al. [28]. We discuss Relationformer in more detail later in the article.

Finally, another way to approach end-to-end learning is to leverage the notion of graph barycenter to define the predicted graph. Relying on the Implicit Loss Embedding (ILE) property of surrogate regression, Brogat-Motte et al. [9] have exemplified this idea by exploiting an Optimal Transport loss, the Fused Gromov-Wasserstein (FGW) distance [45] for which barycenters can be computed efficiently [32, 44]. They proposed two variants, a non-parametric kernel-based one and a neural network-based one, referred to as FGW-Bary and FGW-BaryNN, respectively. However, to calculate the barycenter, the size must be known upstream, leaving the challenge of arbitrary size unresolved. In addition, prediction accuracy is highly dependent on the expressiveness of the barycenter, i.e. the nature and number of graph templates, resulting in high training and inference costs.

In contrast to existing works, our goal is to address the problem of supervised graph prediction in an end-to-end fashion, for different types of input modalities and for output graphs whose size and node ordering can be arbitrary.

**Main contributions**   This paper presents Any2Graph, a versatile framework for end-to-end SGP. Any2Graph leverages a novel, fully differentiable, OT-based loss that satisfies all the previously mentioned properties, i.e., size agnostic and invariant to node permutation. In addition, the encoder part of Any2Graph allows us to leverage inputs of various types, such as images or sets of tokens. We complete our framework with a novel challenging synthetic dataset which we demonstrate to be suited for benchmarking SGP models.

The rest of the paper is organized as follows. After a reminder and a discussion about the relation between graph matching and optimal transport (Section 2), we introduce in Section 3, a *size-agnostic* graph representation and an associated *differentiable* and *node permutation invariant* loss. This loss, denoted as **Partially Masked Fused Gromov Wasserstein** (PMFGW) is a novel and necessary adaptation of the FGW distance [45]. This loss is then integrated into Any2Graph, an end-to-end learning framework depicted in Figure 1 and presented in Section 4. We express the whole framework objective as an ERM problem and highlight the adaptations necessary for extending existing deep learning architectures [36] to more general input modalities.

Section 5, presents a thorough empirical study of Any2Graph on various datasets. We evaluate our approach on four real-world problems with different input modalities as well as *Coloring*, a novel synthetic dataset. As none of the existing approaches could cover the range of input modalities, nor

scale to very large-sized datasets, we adapted them for the purpose of fair comparison. The numerical results showcase the state-of-the-art performances of the proposed method in terms of prediction accuracy and ability to retrieve the right size of target graphs as well as computational efficiency.

## 2 Background on graph matching and optimal transport

**Graph representation and notations**   An attributed graph $g$ with $m$ nodes can be represented by a tuple $(\mathbf{F}, \mathbf{A})$ where $\mathbf{F} = [\mathbf{f}_1, \dots, \mathbf{f}_m]^\top \in \mathbb{R}^{m \times d}$ encodes node features with $\mathbf{f}_i \in \mathbb{R}^d$ labeling each node indexed by $i$, $\mathbf{A} \in \mathbb{R}^{m \times m}$ is a symmetric pairwise distance matrix that describes the graph relationships between the nodes such as the adjacency matrix or the shortest path matrix. Further, we denote $\mathcal{G}_m$ the set of attributed graphs of $m$ nodes and $\mathcal{G} = \bigcup_{m=1}^{M} \mathcal{G}_m$, the set of attributed graphs of size up to $M$, where the size refers to the number of nodes in a graph and the largest size $M$ is an important hyperparameter. In the following, $\mathbf{1}_m \in \mathbb{R}^m$ is the all one vector and we denote $\sigma_m = \{\mathbf{P} \in \{0,1\}^{m \times m} \mid \mathbf{P}\mathbf{1}_m = \mathbf{1}_m, \mathbf{P}^T \mathbf{1}_m = \mathbf{1}_m\}$ the set of permutation matrices.

**Graph Isomorphism**   Two graphs $g_1 = (\mathbf{F}_1, \mathbf{A}_1), g_2 = (\mathbf{F}_2, \mathbf{A}_2) \in \mathcal{G}_m$ are said to be isomorphic whenever there exists $\mathbf{P} \in \sigma_m$ such that $(\mathbf{F}_1, \mathbf{A}_1) = (\mathbf{P}\mathbf{F}_2, \mathbf{P}\mathbf{A}_2\mathbf{P}^T)$, in which case we denote $g_1 \sim g_2$. In this work, we consider all graphs to be unordered, meaning that all operations should be invariant by Graph Isomorphism (GI).

**Comparing graphs of the same size**   Designing a discrepancy to compare graphs is challenging, for instance, even for two graphs of the same size $\hat{g} = (\hat{\mathbf{F}}, \hat{\mathbf{A}})$, $g = (\mathbf{F}, \mathbf{A})$, one cannot simply compute a point-wise comparison as it would not satisfy GI invariance. A solution is to solve a Graph Matching (GM) problem, i.e., to find the optimal matching between the graphs and compute the pairwise errors between matched nodes and edges. This problem can be written as the following

$$\mathrm{GM}(\hat{g}, g) = \min_{\mathbf{P} \in \sigma_m} \sum_{i,j=1}^{m} \mathbf{P}_{i,j} \ell_F(\hat{\mathbf{f}}_i, \mathbf{f}_j) + \sum_{i,j,k,l=1}^{m} \mathbf{P}_{i,j} \mathbf{P}_{k,l} \ell_A(\hat{A}_{i,k}, A_{j,l}). \qquad (1)$$

In particular, with the proper choice of ground metrics $\ell_f$ and $\ell_A$, this is equivalent to the popular Graph Edit Distance (GED) [33]. The minimization problem however is a Quadratic Assignment Problem (QAP) which is known to be one of the most difficult problems in the NP-Hard class [27]. To mitigate this computational complexity, Aflalo et al. [2] suggested to replace the space of permutation matrices with a convex relaxation. The Birkhoff polytope (doubly stochastic matrices) $\pi_m = \{\mathbf{T} \in [0,1]^{m \times m} \mid \mathbf{T}\mathbb{1}_m = \mathbb{1}_m, \mathbf{T}^T \mathbb{1}_m = \mathbb{1}_m\}$ is the tightest of those relaxations as it is exactly the convex hull of $\sigma_m$ which makes it a suitable choice [21]. Interestingly, the resulting metric is known in OT [46] field as a special case of the (Fused) Gromov-Wasserstein (FGW) distance proposed by [29].

$$\mathrm{FGW}(\hat{g}, g) = \min_{\mathbf{T} \in \pi_m} \sum_{i,j=1}^{m} \mathbf{T}_{i,j} \ell_F(\hat{\mathbf{f}}_i, \mathbf{f}_j) + \sum_{i,j,k,l=1}^{m} \mathbf{T}_{i,j} \mathbf{T}_{k,l} \ell_A(\hat{A}_{i,k}, A_{j,l}) \qquad (2)$$

However, these two points of view differ in their interpretation of the FGW metric. From the GM perspective, FGW is cast as an approximation of the original problem, and the optimal transport plan is typically projected back to the space of permutation via Hungarian Matching [47]. From the OT perspective, FGW is used as a metric between distributions with interesting topological properties [44]. This raises the question of the tightness of the relaxation between GM and FGW. In the linear case, i.e., when $\ell_A = 0$, the relaxation is tight and this phenomenon is known in the OT literature as the equivalence between Monge and Kantorovitch formulation [31]. The quadratic case, however, is much more complex, and sufficient conditions under which the tightness holds have been studied in both fields [1, 35].

As seen above, both OT and GM perspectives offer ways to characterize the same objects. In the remainder of this paper, we adopt the OT terminology, e.g., we use the term *transport plan* in place of *doubly stochastic matrix*. We provide a quantitative analysis of the effect of the relaxation in F.2.

**Numerical solver**   Computing the FGW distance requires solving the optimization problem presented in Equation (2) whose objective rewrites $\langle \mathbf{T}, \mathbf{U} \rangle + \langle \mathbf{T}, \mathbf{L} \otimes \mathbf{T} \rangle$ where $\mathbf{U}$ is a fixed matrix, $\mathbf{L}$ a

fixed tensor and $\otimes$ the tensor matrix product. A standard way of solving this problem [47] is to use a conditional gradient (CG) algorithm which iteratively solves a linearization of the problem. Each step of the algorithm requires solving a linear OT/Matching problem of cost $\langle \mathbf{T}, \mathbf{C}^{(k)} \rangle$ where the linear cost $\mathbf{C}^{(k)} = \mathbf{U} + \mathbf{L} \otimes \mathbf{T}^{(k)}$ is updated at each iteration. The linear problem can be solved with a Hungarian solver with cost $\mathcal{O}(M^3)$ while the overall complexity of computing the tensor product $\mathbf{L} \otimes \mathbf{T}^{(k)}$ is theoretically $\mathcal{O}(M^4)$. Fortunately, this bottleneck can be avoided thanks to a $\mathcal{O}(M^3)$ factorization proposed originally by Peyré et al. [32].

**Comparing graphs of arbitrary size** The metrics defined above cannot directly be used to compare graphs of different sizes. To overcome this problem, Vayer et al. [45] proposed a more general formulation that fully leverages OT to model weights on graph nodes and can be used to compare graphs of different sizes as long as they have the same total mass. However, this approach raises specific issues. In scenarios where masses are uniform, nodes in larger graphs receive lower mass which might not be suitable for practical applications. Conversely, employing non-uniform masses complicates interpretation, as decoding a discrete object from a weighted one becomes less straightforward. Those issues can be mitigated by leveraging Unbalanced Optimal Transport (UOT) [40], which relaxes marginal constraints, allowing for different total masses in the graphs. Unfortunately, UOT introduces several additional regularization parameters that are difficult to tune, especially in scenarios like SGP, where model predictions exhibit wide variability during training.

Another close line of work is Partial Matching (PM) [13], which consists in matching a small graph $g$ to a subgraph of the larger graph $\hat{g}$. In practice, this can be done by adding dummy nodes to $g$ through some padding operator $\mathcal{P}$ after which one can directly compute $\text{PM}(\hat{g}, g) = \text{GM}(\hat{g}, \mathcal{P}(g))$ [21]. However, PM is not suited to train a model as the learned model would only be able to predict a graph that includes the target graph. Partial Matching and its relationship with our proposed loss is discussed in more detail in Appendix B.3.

## 3   Optimal Transport loss for Supervised Graph Prediction

**A size-agnostic representation for graphs** Our first step toward building an end-to-end SGP framework is to introduce a space $\hat{\mathcal{Y}}$ to represent any graph of size up to $M$.

$$\hat{\mathcal{Y}} = \{y = (\mathbf{h}, \mathbf{F}, \mathbf{A}) \mid \mathbf{h} \in [0,1]^M, \mathbf{F} \in \mathbb{R}^{M \times d}, \mathbf{A} \in [0,1]^{M \times M}\}. \tag{3}$$

We refer to the elements of $\hat{\mathcal{Y}}$ as continuous graphs, in opposition with discrete graphs of $\mathcal{G}$. Here $h_i$ (resp. $A_{i,j}$) should be interpreted as the probability of the existence of node $i$ (resp. edge $[i, j]$ ). Any graph of $\mathcal{G}$ can be embedded into $\hat{\mathcal{Y}}$ with a padding operator $\mathcal{P}$ defined as

$$\mathcal{P}(g) = \left( \begin{pmatrix} \mathbf{1}_m \\ \mathbf{0}_{M-m} \end{pmatrix}, \begin{pmatrix} \mathbf{F}_m \\ \mathbf{0}_{M-m} \end{pmatrix}, \begin{pmatrix} \mathbf{A}_m & \mathbf{0}_{m, M-m} \\ \mathbf{0}_{M-m,m}^T & \mathbf{0}_{M-m,M-m} \end{pmatrix} \right), \text{ for } g = (\mathbf{F}_m, \mathbf{A}_m) \in \mathcal{G}_m. \tag{4}$$

We denote $\mathcal{Y} = \mathcal{P}(\mathcal{G}) \subset \hat{\mathcal{Y}}$ the space of padded graphs. For any padded graph in $\mathcal{Y}$, the padding operator can be inverted to recover a discrete graph $\mathcal{P}^{-1} : \mathcal{Y} \mapsto \mathcal{G}$. Besides, any continuous graph $\hat{y} \in \hat{\mathcal{Y}}$ can be projected back to padded graphs $\mathcal{Y}$ by a threshold operator $\mathcal{T} : \hat{\mathcal{Y}} \mapsto \mathcal{Y}$. Note that $\hat{\mathcal{Y}}$ is **convex** and of **fixed dimension** which makes it ideal for parametrization with a neural network. Hence, the core idea of our work is to use a neural network to make a prediction $\hat{y} \in \hat{\mathcal{Y}}$ and to compare it to a target $g \in \mathcal{G}$ through some loss $\ell(\hat{y}, \mathcal{P}(g))$. This calls for the design of an asymmetric loss $\ell : \hat{\mathcal{Y}} \times \mathcal{Y} \mapsto \mathbb{R}_+$.

**An Asymmetric loss for SGP** The Partially Masked Fused Gromov Wasserstein (PMFGW) is a loss between a padded target graph $\mathcal{P}(g) = (\mathbf{h}, \mathbf{F}, \mathbf{A}) \in \mathcal{Y}$ with real size $m = \|\mathbf{h}\|_1 \leq M$ and a continuous prediction $\hat{y} = (\hat{\mathbf{h}}, \hat{\mathbf{F}}, \hat{\mathbf{A}}) \in \hat{\mathcal{Y}}$. We define PMFGW$(\hat{y}, \mathcal{P}(g))$ as:

$$\min_{\mathbf{T} \in \pi_M} \frac{\alpha_{\mathrm{h}}}{M} \sum_{i,j} T_{i,j} \ell_h(\hat{h}_i, h_j) + \frac{\alpha_{\mathrm{f}}}{m} \sum_{i,j} T_{i,j} \ell_f(\hat{\mathbf{f}}_i, \mathbf{f}_j) h_j + \frac{\alpha_{\mathrm{A}}}{m^2} \sum_{i,j,k,l} T_{i,j} T_{k,l} \ell_A(\hat{A}_{i,k}, A_{j,l}) h_j h_l. \tag{5}$$

Let us decompose this loss function to understand the extend to which it simultaneously takes into account each property of the graph. The first term ensures that the padding of a node is well predicted. In particular, this requires the model to predict correctly the number of nodes in the target graph. The

second term ensures that the features of all non-padding nodes ($h_i = 1$) are well predicted. Similarly, the last term ensures that the pairwise relationships between non-padded nodes ($h_i = h_j = 1$) are well predicted. The normalizations in front of the sums ensure that each term is a weighted average of its internal losses as $\sum T_{i,j} = M$, $\sum T_{i,j} h_j = m$ and $\sum T_{i,j} T_{k,l} h_j h_l = m^2$. Finally $\boldsymbol{\alpha} = [\alpha_h, \alpha_f, \alpha_A] \in \Delta_3$ is a triplet of hyperparameters on the simplex balancing the relative scale of the different terms. For $\ell_A$ and $\ell_h$ we use the cross-entropy between the predicted value after a sigmoid and the actual binary value in the target. This is equivalent to a logistic regression loss after the OT plan has matched the nodes. For $\ell_f$ we use the squared $\ell_2$ or the cross-entropy loss when the node features are continuous or discrete, respectively.

A key feature of this loss is its flexibility. Not only other ground losses can be considered but it is also straightforward to introduce richer spectral representations of the graph [4]. For instance, in Section 5, we explore the benefits of leveraging a diffused version of the nodes features.

Finally, PMFGW translates all the good properties of FGW to the new size-agnostic representation.

**Proposition 1** (Complexity). *The objective of the inner optimization can be evaluated in $\mathcal{O}(M^3)$.*

**Proposition 2** (GI Invariance). *If $\hat{y} \sim \hat{y}'$ and $g \sim g'$ then PMFGW($\hat{y}, \mathcal{P}(g)$) = PMFGW($\hat{y}', \mathcal{P}(g')$).*

**Proposition 3** (Positivity). *PMFGW($\hat{y}, \mathcal{P}(g)$) $\geq 0$ with equality if and only if $\hat{y} \sim \mathcal{P}(g)$.*

See Appendix A for a toy example illustrating the behavior of the loss and Appendix B.1 and B.2 for formal statements and proofs of Proposition 1, 2 and 3.

**Relation to existing metrics**   PMFGW is an asymmetric extension of FGW [44] suited for comparing a continuous predicted graph with a padded target. The extension is achieved by adding (1) a novel term to quantify the prediction of node padding, and (2) the partial masking of the components of the second and third terms to reflect padding. It should be noted that in contrast to what is usually done in OT, the node masking vectors ($\mathbf{h}$ and $\hat{\mathbf{h}}$) are not used as a marginal distribution but directly integrated into the loss. In that sense, the additional node masking term is very similar to the one of OTL$_p$ [39] that proposed to use uniform marginal weight and move the part that measures the similarity between the distribution weights in an additional linear term. However, OTL$_p$ is restricted to linear OT problems and does not use the marginal distributions as a masking for other terms as in PMFGW.

PMFGW also relates to Partial GM/GW Chapel et al. [13] as both metrics compare graphs by padding the smallest one with zero-cost dummy nodes. The critical difference lies in the new vector $\hat{\mathbf{h}}$ which predicts which sub-graphs are activated, i.e., should be matched to the target. The exact relationship between Partial Fused Gromov Wasserstein (PFGW) and PMFGW is summarized below

**Proposition 4.** *If $l_h$ is set to a constant value, PMFGW is equal to PFGW (up to a constant).*

*Remark* 1. In that case, the vector $\hat{\mathbf{h}}$ disappears from the loss and cannot be trained. In particular, this would prevent the model from learning to predict the size of the target graph.

The formal definition of PFGW and the proof of Proposition 4 are provided in Appendix B.3.

## 4   Any2Graph: a framework for end-to-end SGP

### 4.1   Any2Graph problem formulation

The goal of Supervised Graph Prediction (SGP) is to learn a function $f : \mathcal{X} \to \mathcal{G}$ using the training samples $\{(x_i, g_i)\}_{i=1}^n \in (\mathcal{X} \times \mathcal{G})^n$. In Any2Graph, we relax the output space and learn a function $\hat{f} : \mathcal{X} \to \hat{\mathcal{Y}}$ that predicts a continuous graph $\hat{y} := f(x)$ as defined in the previous section. Assuming $\hat{f}$ is a parametric model (in this work, a deep neural network) completely determined by a parameter $\theta$, the Any2Graph objective writes as the following empirical risk minimization problem:

$$\min_{\theta} \quad \frac{1}{n} \sum_{i=1}^n \text{PMFGW}(\hat{f}_\theta(x_i), \mathcal{P}(g_i)). \tag{6}$$

At inference time, we recover a discrete prediction by a straightforward decoding $f(x) = \mathcal{P}^{-1} \circ \mathcal{T}(\hat{y})$, where $\mathcal{T}$ is the thresholding operator with threshold $1/2$ on the edges and nodes and $\mathcal{P}^{-1}$ is the inverse

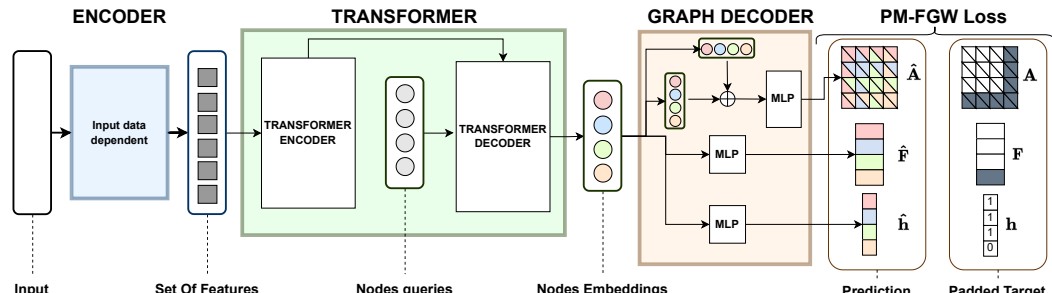

Figure 1: Illustration of the architecture for a target graph of size 3 and $M = 4$.

padding defined in the previous section. In other words, the full decoding pipeline $\mathcal{P}^{-1} \circ \mathcal{T}$ removes the nodes $i$ (resp. edges $(i, j)$) whose predicted probability is smaller than $1/2$ i.e. $\hat{h}_i < 1/2$ (resp. $\hat{A}_{i,j} < 1/2$)). Unlike surrogate regression methods, this decoding step is very efficient.

### 4.2 Neural network architecture

The model $\hat{f}_\theta : \mathcal{X} \to \hat{\mathcal{Y}}$ (left part of Figure 1) is composed of three modules , namely the **encoder** that extracts features from the input, the **transformer** that convert these features into $M$ nodes embeddings, that are expected to capture both feature and structure information, and the **graph decoder** that predicts the properties of our output graph, i.e., $(\hat{\mathbf{h}}, \hat{\mathbf{F}}, \hat{\mathbf{A}})$. As we will discuss later, the proposed architecture draws heavily on that of Relationformer [36] since the latter has been shown to yield to state-of-the-art results on the Image2Graph task.

**Encoder** The encoder extracts $k$ feature vectors in $\mathbb{R}^{d_e}$ from the input. Note that $k$ is not fixed a priori and can depend on the input (for instance sequence length in case of text input). This is critical for encoding structures as complex as graphs and the subsequent transformer is particularly apt at treating this kind of representation. By properly designing the encoder, we can accommodate different types of input data. In Appendix C, we describe how to handle images, text, graphs, and vectors and provide general guidelines to address other input modalities.

**Transformer** This module takes as input a set of feature vectors and outputs a fixed number of $M$ node embeddings. This resembles the approach taken in machine translation, and we used an architecture based on a stack of transformer encoder-decoders, akin to Shit et al. [36].

**Graph decoder** This module decodes a graph from the set of node embeddings $\mathbf{Z} = [\mathbf{z}_1, \ldots, \mathbf{z}_M]^T$ using the following equation:

$$\hat{h}_i = \sigma(\text{MLP}_m(\mathbf{z}_i)), \quad \hat{F}_i = \text{MLP}_f(\mathbf{z}_i), \quad \hat{A}_{i,j} = \sigma(\text{MLP}_s^2(\text{MLP}_s^1(\mathbf{z}_i) + \text{MLP}_s^1(\mathbf{z}_j))) \tag{7}$$

where $\sigma$ is the sigmoid function and $\text{MLP}_m$, $\text{MLP}_f$, $\text{MLP}_s^k$ are multi-layer perceptrons heads corresponding to each component of the graph (mask, features, structure). The adjacency matrix is expected to be symmetric which motivate us to parameterize it as suggested by Zaheer et al. [56].

**Positioning with Relationformer** As discussed above, the architecture is similar to the one proposed in Relationformer [36], with two modifications: (1) we use a symmetric operation with a sum to compute the adjacency matrix while Relationformer uses a concatenation that is not symmetric; (2) we investigate more general encoders to enable graph prediction from data other than images. However, as stated in the previous section, the main originality of our framework lies in the design of the PMFGW loss. Interestingly Relationformer uses a loss that presents similarities with FGW but where the matching is done on the node features only, before computing a quadratic-linear loss similar to PMFGW. In other words, they solve a bi-level optimization problem, where the plan is computed on only part of the information, leading to potentially suboptimal results on heterophilic graphs as demonstrated in the next section.

# 5 Numerical experiments

This section is organized as follows. First, we describe the experimental setting (5.1) including baselines. Next, we showcase the state-of-the-art performances of Any2Graph for a wide range of metrics and datasets (5.2). Finally, we provide an empirical analysis of the key hyperparameters of Any2Graph (5.3). The code for Any2Graph and all the experiments will be released on GitHub.

## 5.1 Experimental setting

**Datasets** We consider 5 datasets thus covering a wide spectrum of different input modalities, graph types, and sizes. The first one, *Coloring*, is a new synthetic dataset that we proposed, inspired by the four-color theorem. The input is a noisy image partitioned into regions of colors and the goal is to predict the graph representing the regions as nodes (4 color classes) and their connectivity in the image. An example is provided in Figure 5 and more details are in Appendix D. Then, we consider four real-world benchmarks. *Toulouse* [5] is Sat2Graph datasets where the goal is to extract the road network from binarized satellite images of a city. *USCities* is also a Sat2Graph dataset but features larger and more convoluted graphs. Note that we leave aside the more complex RGB version of *USCities* as it was shown to require complex multi-level attention architecture [36], which is beyond the scope of this paper. Finally, following Ucak et al. [42], we address the Fingerprint2Graph task where the goal is to reconstruct a molecule from its fingerprint representation (list of tokens). We consider two widely different datasets for this tasks: *QM9* [50], a scarce dataset of tiny molecules (up to 9 nodes) and *GBD13*Blum and Reymond [7], a large dataset [2] featuring molecules with up to 13 heavy atoms. Additional details concerning the datasets (e.g. dataset size, number of edges, number of nodes) are provided in Appendix E.1.

**Compared methods** We compare Any2Graph, to our direct end-to-end competitor Relationformer [36] that has shown to be the state-of-the-art method for Image2Graph. For a fair comparison, we use the same architecture (presented in Figure 1) for both approaches so that the only difference is the loss. We conjecture that Any2Graph and Relationformer might benefit from feature diffusion, that is replacing the node feature vector $\mathbf{F}$ by the concatenation $[\mathbf{F}, \mathbf{AF}]$ before training. We denote by "+FD" the addition of feature diffusion before training. Moreover, we also compare with a surrogate regression approach (FGW-Bary) based on FGW barycenters [9]. We test both the end-to-end parametric variant, FGWBary-NN, where weight functions $\alpha_k$, as well as $K = 10$ templates, are learned by a neural network and the non-parametric variant, FGWBary-ILE, where the templates are training samples and $\alpha_k$ are learned by sketched kernel ridge regression [54, 18] with gaussian kernel. Both have been implemented using the codes provided by Brogat-Motte et al. [9], modified to incorporate sketching. Hyperparameters regarding architectures and optimization are provided in Appendix E.2.

**Performance metrics** The heterogeneity of our datasets, calls for task-agnostic metrics focusing on different fine-grained levels of the graph. At the graph level, we report the PMFGW loss between continuous prediction $\hat{y}$ and padded target $\mathcal{P}(g)$ and the graph edit distance Gao et al. [19] between predicted graph $\mathcal{P}^{-1}\mathcal{T}\hat{y}$ and target $g$. We also report the Graph Isomorphism Accuracy (GI ACC), a metric that computes the proportion of perfectly predicted graphs. At the edge level, we treat the adjacency matrix prediction as a binary prediction problem and report the Precision and Recall. Finally, at the node level, we report NODE, the fraction of well-predicted node features, and SIZE a metric reporting the accuracy of the predicted number of nodes. See Appendix E.3 for more details.

## 5.2 Comparison with existing SGP methods on diverse modalities

**Prediction Performances** Table 1 shows the performances of the different methods on the five datasets. First, we observe that Any2Graph achieves state-of-the-art performances for all datasets and graph level metrics. On the Sat2Graph tasks (*Toulouse* and *USCities*) we note that Relationformer performs very close to Any2Graph. In fact, the features (2D positions) are enough to uniquely identify the nodes, making Relationformer's Hungarian matching sufficient. Moreover, both methods highly benefit from feature diffusion on Fingerprint2Graph tasks which we discuss further in Appendix F. Note that both barycenter methods struggle on *Toulouse*, possibly due to a lack of expressivity. On

---

[2]For computational purposes we use the 'ABCDEFGH' subset (more than 1.3 millions molecules) .

Table 1: Graph level, edge level, and node level metrics reported on test for the different models and datasets. * denotes methods that use the actual size of the graph at inference time, hence the performance reported is a non-realistic upper bound. We where not able to train FGWBary on all dataset due to the prohibitive cost of barycenter computations. N.A. stands for not applicable.

| Dataset | Model | Graph Level | | | Edge Level | | Node Level Acc. | |
|---|---|---|---|---|---|---|---|---|
| | | Edit Dist. ↓ | GI Acc. ↑ | PMFGW ↓ | Prec. ↑ | Rec. ↑ | Node ↑ | Size ↑ |
| Coloring | FGWBary-NN* | 6.73 | 1.00 | 0.91 | 75.19 | 84.99 | 77.58 | N.A. |
| | FGWBary-ILE* | 7.60 | 0.90 | 0.93 | 72.17 | 83.81 | 79.15 | N.A. |
| | Relationformer | 5.47 | 18.14 | 0.32 | 80.39 | 86.34 | 92.68 | 99.32 |
| | Any2Graph | **0.20** | **85.20** | **0.03** | **99.15** | **99.37** | **99.95** | **99.50** |
| Toulouse | FGWBary-NN* | 8.11 | 0.00 | 1.15 | 84.09 | 79.68 | 10.10 | N.A. |
| | FGWBary-ILE* | 9.00 | 0.00 | 1.21 | 72.52 | 56.30 | 1.62 | N.A. |
| | Relationformer | **0.13** | 93.28 | **0.02** | 99.25 | 99.24 | 99.25 | 98.30 |
| | Any2Graph | **0.13** | **93.62** | **0.02** | **99.34** | **99.26** | **99.39** | **98.81** |
| USCities | Relationformer | 2.09 | 55.00 | 0.13 | **92.96** | 87.98 | 95.18 | **79.80** |
| | Any2Graph | **1.86** | **58.10** | **0.12** | 92.91 | **90.85** | **95.70** | 78.95 |
| QM9 | FGWBary-NN* | 5.55 | 1.00 | 0.96 | 87.81 | 70.78 | 78.62 | N.A. |
| | FGWBary-ILE* | 3.54 | 7.10 | 0.59 | 80.40 | 75.14 | 91.47 | N.A. |
| | FGWBary-ILE* + FD | 2.84 | 28.95 | 0.28 | 82.96 | 79.76 | 92.99 | N.A. |
| | Relationformer | 9.15 | 0.05 | 0.48 | 21.42 | 4.77 | 99.28 | 91.80 |
| | Relationformer + FD | 3.80 | 9.95 | 0.22 | 86.07 | 73.31 | 99.34 | **96.0** |
| | Any2Graph | 3.44 | 7.50 | 0.21 | 86.21 | 77.27 | 99.26 | 93.65 |
| | Any2Graph + FD | **2.13** | **29.85** | **0.14** | **90.19** | **88.08** | **99.77** | 95.45 |
| GDB13 | Relationformer | 11.40 | 0.00 | 0.43 | 81.96 | 31.49 | 97.77 | 97.45 |
| | Relationformer + FD | 8.83 | 0.01 | 0.29 | 84.14 | 55.89 | 97.57 | **98.65** |
| | Any2Graph | 7.45 | 0.05 | 0.22 | 87.20 | 60.41 | 99.41 | 96.15 |
| | Any2Graph + FD | **3.63** | **16.25** | **0.11** | **90.83** | **84.86** | **99.80** | 98.15 |

*QM9* on the other hand, FGW-Bary-ILE performs close to Any2Graph but is still outperformed, even more so if we add feature diffusion. It should be stressed that FGW-Bary-ILE is placed here under the most favorable conditions possible, in particular with the use of a SOTA kernel and being given the ground truth number of nodes. Finally, on *GDB13*, the most challenging dataset, Any2Graph strongly outperforms all competitors. To summarize this part, we observe that Any2Graph and its variant Any2Graph+FD are consistently better by a large margin on three out of five benchmarks and tied for first place with RelationFormer on the remaining ones.

**Computational Performances**  Table 2 shows the cost of training and inference of all methods, expressed as the number of graphs processed per second. All values are computed on NVIDIA V100/Intel Xeon E5-2660. The heavy cost of barycenters computation in FGWBary makes it several orders of magnitude slower than Any2Graph. We note that Relationformer is faster at learning time because it avoids the need to solve a QAP. Overall, our proposed approach offers the best of both worlds, achieving SOTA prediction performance at all levels of the graph, at a very low computational inference cost.

Table 2: Computational speed for the different methods in terms of graphs per second on *QM9*. Training performance is not provided for FGWBary-ILE as the closed-form expression is computed at once on CPU.

| Method | Train. | Pred. |
|---|---|---|
| FGWBary-ILE ($K=25$) | N.A. | 1 |
| FGWBary-NN ($K=10$) | 10 | 10 |
| Relationformer | 2k | 10k |
| Any2Graph | 1k | 10k |

**Qualitative Performances**  We display a sample of the graph predictions performed by the models in Figure 5 (left) along with a larger collection in appendix G. We observe that not only Any2Graph can adapt to different input modalities, but also to the variety of target graphs at hand. For instance, *Coloring* graphs tend to be denser, while Sat2Graph maps can contain multiple connected components and Fingerprint2Graph molecules exhibit unique substructures such as loops.

## 5.3 Empirical study of Any2Graph properties

**Sensitivity to dataset size**  In figure 5, we provide the test edit distance for different numbers of training samples in the explored datasets. Interestingly, we observe that a performance plateau is reached for all datasets. We also observe that *Coloring/Toulouse* are simpler than *USCities/QM9* (in

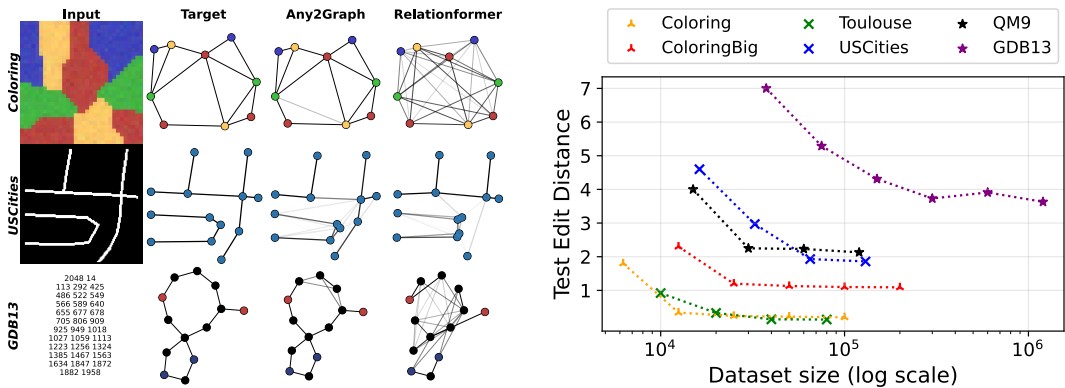

Figure 5: (Left): a sample of predictions made by Any2Graph and Relationformer for a given input and ground truth target. Many more are provided in G. (Right): we truncate the train datasets to provide an overview of Any2Graph training curves (test performances against train set size).

terms of best edit distance) and illustrate the flexibility of *Coloring* by bridging this complexity gap with the more challenging variation described in D.

**Scalability to larger graphs** As stated in property 1, each iteration of the PMFGW solver scales with $\mathcal{O}(M^3)$. Denoting $k(M)$ the average number of iterations required for convergence, this means that the actual cost of computing the loss scales with $\mathcal{O}\left(k(M)M^3\right)$. We provide an empirical estimation of $k(M)$ in figure 2 which we obtain by computing $\mathrm{PMFGW}(\mathcal{P}(g_1), \mathcal{P}(g_2))$ for pairs of graphs $g_1, g_2$ sampled from the *Coloring* dataset. We observe that $k(M)$ seems linear but can be made sub-linear using feature diffusion (FD). Still, the cubic cost prevents Any2Graph from scaling beyond a few tens of nodes.

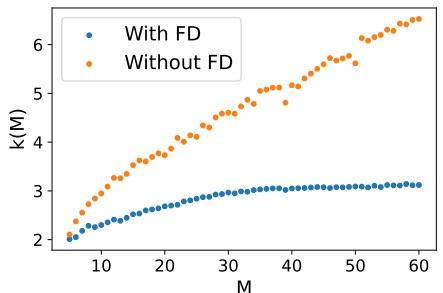

Figure 2: Average number of solver iterations required for computing PMFGW loss.

**Choice of maximal graph size** $M$ The default value of $M$ is the size of the largest graph in the train set. We explore whether or not overparametrizing the model with higher values bring substantial benefits. To this end, we train our model on *Coloring* for $M$ between 10 (default value) and 25 and report the (test) edit distance. To quantify the effective number of nodes used by the model, we also record the number of active nodes, i.e., that are masked less than 99% of the time (see Figure 3). Interestingly, we observe that performances are robust w.r.t. the choice of $M$ which can be explained by the number of useful nodes reaching a plateau. This suggests the model automatically learns the number of nodes it needs to achieve the required expressiveness.

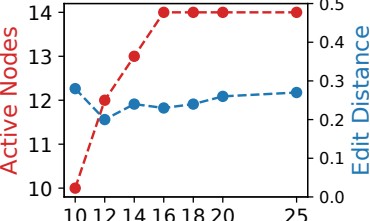

Figure 3: Effect of $M$ on test edit distance and number of active nodes for *Coloring*.

**Sensitivity to the weights $\alpha$** We investigate the sensitivity of the proposed loss to the triplet of weight hyperparameters $\alpha$. To this end, we train our model on *Coloring* for different values $\alpha$ on the simplex and report the (test) edit distance on Figure 4. We observe that the performance is optimal for uniform $\alpha$ and robust to other choices as long as there is not too much weight on the structure loss term (corner $\alpha_A = 1$). Indeed, the quadratic term of the loss being the hardest, putting too much emphasis on it might slow down the training. This explains the efficiency of feature diffusion, as it moves parts of the structure prediction to the linear term. Further evidence backing this intuitive explanation is provided in F.1.

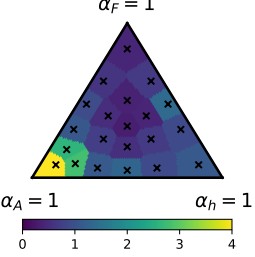

Figure 4: Effect of $\alpha$ on the test edit distance.

# 6 Conclusion and limitations

We present Any2Graph, a novel deep learning approach to Supervised Graph Prediction (SGP) leveraging an original asymmetric Partially-Masked Fused Gromov-Wasserstein loss. To the best of our knowledge, Any2Graph stands as the first end-to-end and versatile framework to consistently achieve state-of-the-art performances across a wide range of graph prediction tasks and input modalities. Notably, we obtain excellent results in both accuracy and computational efficiency. Finally, we illustrate the adaptability of the proposed loss (using diffused features), and the robustness of the method to sensitive parameters such as maximum graph size and weights of the different terms in PMFGW.

The main limitation of Any2Graph is its scalability to graphs of larger size. Considering the tasks at hand, in this paper, we limit ourselves to relatively small graphs of up to 20 nodes.
For the future, we envision two working directions to address this issue. First, given the promising results with feature diffusion, we plan to introduce more tools from spectral graph theory [14, 20, 4] in Any2Graph, e.g., using diffusion on the adjacency matrix to capture higher-order interactions that may occur in large graphs. More generally, this extension could be useful even for smaller graphs. Secondly, we expect that the solver computing the optimal transport plan can be accelerated using approximation. In particular, the entropic regularization [32] might unlock the possibility of fully parallelizing the optimization on a GPU while low-rank OT solvers Scetbon et al. [34] could allow Any2Graph to scale to large output graphs.

# Acknowledgements

The authors thanks Alexis Thual and Quang Huy Tran for providing their insights and code about the Fused Unbalanced Gromov Wasserstein metric. This work received funding from the European Union's Horizon Europe research and innovation programme under grant agreement 101120237 (ELIAS). Views and opinions expressed are however those of the authors only and do not necessarily reflect those of the European Union or European Commission. Neither the European Union nor the granting authority can be held responsible for them. This research was also supported in part by the French National Research Agency (ANR) through the PEPR IA FOUNDRY project (ANR-23-PEIA-0003) and the MATTER project (ANR-23-ERCC-0006-01). The first and second authors respectively received PhD scholarships from Institut Polytechnique de Paris and Hi!Paris.

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

# A  Illustration of PMFGW On A Toy Example

On the one hand, we consider a target graph of size 2, $\mathbf{g} = (\mathbf{F}_2, \mathbf{A}_2)$ where

$$\mathbf{F}_2 = \begin{pmatrix} \mathbf{f}_1 \\ \mathbf{f}_2 \end{pmatrix} ; \mathbf{A}_2 = \begin{pmatrix} 0 & 1 \\ 1 & 0 \end{pmatrix}$$

for some node features $\mathbf{f}_1$ and $\mathbf{f}_2$. For $M = 3$ the padded target is $\mathcal{P}(\mathbf{g}) = (\mathbf{h}, \mathbf{F}, \mathbf{A})$ where

$$\mathbf{h} = \begin{pmatrix} 1 \\ 1 \\ 0 \end{pmatrix} ; \mathbf{F} = \begin{pmatrix} \mathbf{f}_1 \\ \mathbf{f}_2 \\ - \end{pmatrix} ; \mathbf{A} = \begin{pmatrix} 0 & 1 & - \\ 1 & 0 & - \\ - & - & - \end{pmatrix}.$$

On the other hand, we consider a predicted graph $\hat{\mathbf{y}}_{a,h} = (\hat{\mathbf{h}}, \hat{\mathbf{F}}, \hat{\mathbf{A}})$ that has the form

$$\hat{\mathbf{h}} = \begin{pmatrix} 1 \\ h \\ 1-h \end{pmatrix} ; \hat{\mathbf{F}} = \begin{pmatrix} \mathbf{f}_1 \\ \mathbf{f}_2 \\ \mathbf{f}_2 \end{pmatrix} ; \hat{\mathbf{A}} = \begin{pmatrix} 0 & a & 1-a \\ a & 0 & 0 \\ 1-a & 0 & 0 \end{pmatrix}$$

for some $a, h \in [0, 1]$. The loss between the prediction and the (padded) target is

$$\mathcal{L}_{\text{train}}(a, h) = \text{PMFGW}(\hat{\mathbf{y}}_{a,h}, \mathcal{P}_3(\mathbf{g}))$$

We are interested in the landscape of this loss. First of all, it appears that $\hat{\mathbf{y}}_{1,1}$ and $\hat{\mathbf{y}}_{0,0}$ and $\mathcal{P}(\mathbf{g})$ are isomorphic, thus we get two global minima $\mathcal{L}_{\text{train}}(1, 1) = \mathcal{L}_{\text{train}}(0, 0) = 0$. Going into greater detail, it can be shown that for $\ell_h(a, b) = \ell_A(a, b) = (a - b)^2$ we have the following expression

$$\mathcal{L}_{\text{train}}(a, h) = \min\left( (1 - a)^2 + \frac{2}{3}(1 - h)^2 ; a^2 + \frac{2}{3}h^2 \right)$$

and the optimal transport plan is the permutation $(1, 2, 3)$ when $(1 - a)^2 + \frac{2}{3}(1 - h)^2 < a^2 + \frac{2}{3}h^2$ and $(1, 3, 2)$ otherwise. In this toy example, the optimal transport plan is always a permutation.

At inference time, we could similarly be interested in the edit distance between the (discrete) prediction and the target

$$\mathcal{L}_{\text{eval}}(a, h) = \text{ED}(\mathcal{P}_3^{-1}\mathcal{T}(\hat{\mathbf{y}}_{a,h}), \mathbf{g}).$$

Once again, the expression can be computed explicitly

$$\mathcal{L}_{\text{eval}}(a, h) = \mathbb{1}[a < 0.5 \text{ and } h > 0.5] + \mathbb{1}[a > 0.5 \text{ and } h < 0.5]$$

We provide in Figure 6 an illustration of the edit distance and the proposed loss that is clearly a continuous and smoothed version of the edit distance which allows for learning the NN parameters.

# B  Formal Statements And Proofs

In this section, we write

$$\text{PMFGW}(\hat{y}, \mathcal{P}(g)) = \min_{\mathbf{T} \in \pi_M} \sum_{i,j} T_{i,j} \ell_h(\hat{h}_i, h_j) + \sum_{i,j} T_{i,j} \ell_f(\hat{\mathbf{f}}_i, \mathbf{f}_j) h_j + \sum_{i,j,k,l} T_{i,j} T_{k,l} \ell_A(\hat{A}_{i,k}, A_{j,l}) h_j h_l,$$

meaning that we absorb the normalization factors in the ground losses to lighten the notation.

Alternatively, we also consider the matrix formulation:

$$\text{PMFGW}(\hat{y}, \mathcal{P}(g)) = \min_{\mathbf{T} \in \pi_M} \langle \mathbf{T}, \mathbf{C} \rangle + \langle \mathbf{T}, \mathbf{L} \otimes \mathbf{T} \rangle,$$

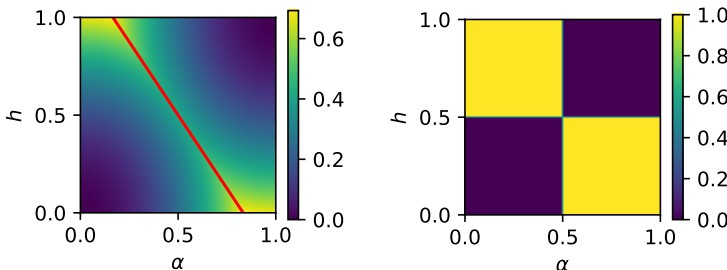

Figure 6: Heatmap of $\mathcal{L}_{\text{train}}$ (left) and $\mathcal{L}_{\text{eval}}$ (right). The red line represents the transition between the regime where the optimal transport plan is the permutation $(1, 2, 3)$ and that where it is $(1, 3, 2)$. In both cases, the optimal transport plan is a permutation.

where $C_{i,j} = \ell_h(\hat{h}_i, h_j) + \ell_f(\hat{\mathbf{f}}_i, \mathbf{f}_j)h_j$, $L_{i,j,k,l} = \ell_A(\hat{A}_{i,k}, A_{j,l})h_j h_l$ and $\otimes$ is the tensor/matrix product.

## B.1 PMFGW fast computation

The following results generalize the Proposition 1 of Peyré et al. [32] so that it can be applied to the computation of PMFGW.

**Proposition 5.** *Assuming that the ground loss than can decomposed as $\ell(a,b) = f_1(a) + f_2(b) - h_1(a)h_2(b)$, for any transport plan $\mathbf{T} \in \mathbb{R}^{n \times m}$ and matrices $\mathbf{A}, \mathbf{W} \in \mathbb{R}^{n \times n}$ and $\mathbf{A}', \mathbf{W}' \in \mathbb{R}^{m \times m}$, then the tensor product of the form*

$$(\mathbf{L} \otimes \mathbf{T})_{i,i'} = \sum_{j,j'} T_{j,j'}\ell(A_{i,j}, A'_{i',j'})W_{i,j}W'_{i',j'}$$

*can be computed as*

$$\mathbf{L} \otimes \mathbf{T} = \mathbf{U}_1\mathbf{T}\mathbf{W}'^T + \mathbf{W}\mathbf{T}\mathbf{U}_2^T - \mathbf{V}_1\mathbf{T}\mathbf{V}_2^T,$$

*where $\mathbf{U}_1 = f_1(\mathbf{A}) \cdot \mathbf{W}$, $\mathbf{U}_2 = f_2(\mathbf{A}') \cdot \mathbf{W}'$, $\mathbf{V}_1 = h_1(\mathbf{A}) \cdot \mathbf{W}$, $\mathbf{V}_2 = h_2(\mathbf{A}') \cdot \mathbf{W}'$ and $[\cdot]$ is the point-wise multiplication.*

*Proof.* Thanks to the decomposition assumption the tensor product can be decomposed into 3 terms:

$$(\mathbf{L} \otimes \mathbf{T})_{i,i'} = \sum_{j,j'} T_{j,j'}f_1(A_{i,j})W_{i,j}W'_{i',j'} + \sum_{j,j'} T_{j,j'}f_2(A'_{i',j'})W_{i,j}W'_{i',j'} - \sum_{j,j'} T_{j,j'}h_1(A_{i,j})h_2(A'_{i',j'})W_{i,j}W'_{i',j'}.$$

$$= \sum_{j} f_1(A_{i,j})W_{i,j}\sum_{j'} T_{j,j'}W'_{i',j'} + \sum_{j'} f_2(A'_{i',j'})W'_{i',j'}\sum_{j} T_{j,j'}W_{i,j} - \sum_{j} h_1(A_{i,j})W_{i,j}\sum_{j'} T_{j,j'}h_2(A'_{i',j'})W'_{i',j'}.$$

Introducing $\mathbf{U}_1, \mathbf{U}_2, \mathbf{V}_1$ and $\mathbf{U}_2$ as defined above, we write:

$$(\mathbf{L} \otimes \mathbf{T})_{i,i'} = \sum_{j}(U_1)_{i,j}\sum_{j'} T_{j,j'}W'_{i',j'} + \sum_{j'}(U_2)_{i',j'}\sum_{j} T_{j,j'}W_{i,j} - \sum_{j}(V_1)_{i,j}\sum_{j'} T_{j,j'}(V_2)_{i',j'},$$

$$= \sum_{j}(U_1)_{i,j}(TW'^T)_{j,i'} + \sum_{j'}(U_2)_{i',j'}(WT)_{i,j'} - \sum_{j}(V_1)_{i,j}(T_{j,j'}V_2^T)_{j,i'},$$

which concludes that $\mathbf{L} \otimes \mathbf{T} = \mathbf{U}_1\mathbf{T}\mathbf{W}'^T + \mathbf{W}\mathbf{T}\mathbf{U}_2^T - \mathbf{V}_1\mathbf{T}\mathbf{V}_2^T$. $\qquad\square$

*Remark* 2 (Computational cost). $\mathbf{U}_1, \mathbf{U}_2, \mathbf{V}_1, \mathbf{V}_2$ can be pre-computed for a cost of $\mathcal{O}(n^2 + m^2)$, after which $\mathbf{L} \otimes \mathbf{T}$ can be computed (for any $\mathbf{T}$) at a cost of $\mathcal{O}(mn^2 + nm^2)$.

*Remark* 3 (Kullback-Leibler divergence decomposition). The Kullback-Leibler divergence $KL(p,q) = q\log\frac{q}{p} + (1-q)\log\frac{(1-q)}{(1-p)}$, which we use as ground loss in our experiments satisfies the required decomposition given $f_1(p) = -\log(p)$, $f_2(q) = q\log(q) + (1-q)\log(1-q)$, $h_1(p) = \log(\frac{1-p}{p})$ and, $h_2(q) = 1-q$

*Remark* 4. The tensor product that appears in PMFGW is a special case of this theorem that corresponds to $n = m = M$, $W_{i,j} = 1$ and $W'_{i',j'} = h_{i'}h_{j'}$. Thus, proposition 1 is a direct corollary.

## B.2 PMFGW divergence properties

First, we provide below a more detailed version of Proposition 2

**Proposition 6** (GI Invariance). *For any $m \leq M$, $\hat{y}, \hat{y}' \in \hat{\mathcal{Y}}$ and $g, g' \in \mathcal{G}_m$, we have that*

$$\hat{y} \sim \hat{y}', g \sim g' \implies PMFGW(\hat{y}, \mathcal{P}(g)) = PMFGW(\hat{y}', \mathcal{P}(g')).$$

*Proof.* We denote $\hat{y} = (\hat{\mathbf{h}}, \hat{\mathbf{F}}, \hat{\mathbf{A}})$ and $\mathcal{P}(g) = (\mathbf{h}, \mathbf{F}, \mathbf{A})$. Since $\hat{y}$ and $\hat{y}'$ are isomorphic, there exist a permutation $\mathbf{P} \in \sigma_M$ such that $\hat{y}' = (\mathbf{P}\hat{\mathbf{h}}, \mathbf{P}\hat{\mathbf{F}}, \mathbf{P}\hat{\mathbf{A}}\mathbf{P}^T)$. Moreover, the fact that $g$ and $g'$ are isomorphic implies that $\mathcal{P}(g)$ and $\mathcal{P}(g')$ are isomorphic as well, thus there exist a permutation $\mathbf{Q} \in \sigma_M$ such that $\mathcal{P}(g') = (\mathbf{Q}\mathbf{h}, \mathbf{Q}\mathbf{F}, \mathbf{Q}\mathbf{A}\mathbf{Q}^T)$. Plugging into the PMFGW objective we get

$$\begin{aligned}
\text{PMFGW}(\hat{y}', \mathcal{P}(g')) &= \min_{\mathbf{T} \in \pi_M} \sum_{i,j} T_{i,j}\ell_h((\mathbf{P}\hat{\mathbf{h}})_i, (\mathbf{Q}\mathbf{h})_j) + \sum_{i,j} T_{i,j}\ell_f((\mathbf{P}\hat{\mathbf{F}})_i, (\mathbf{Q}\mathbf{F})_j)(\mathbf{Q}\mathbf{h})_j \\
&\quad + \sum_{i,j,k,l} T_{i,j}T_{k,l}\ell_A((\mathbf{P}\hat{\mathbf{A}}\mathbf{P}^T)_{i,k}, (\mathbf{Q}\mathbf{A}\mathbf{Q}^T)_{j,l})(\mathbf{Q}\mathbf{h})_j(\mathbf{Q}\mathbf{h})_l \\
&= \min_{\mathbf{T} \in \pi_M} \sum_{i,j} (\mathbf{P}^T\mathbf{T}\mathbf{Q})_{i,j}\ell_h(\hat{\mathbf{h}}_i, \mathbf{h}_j) + \sum_{i,j} (\mathbf{P}^T\mathbf{T}\mathbf{Q})_{i,j}\ell_f(\hat{\mathbf{f}}_i, \mathbf{f}_j)h_j \\
&\quad + \sum_{i,j,k,l} (\mathbf{P}^T\mathbf{T}\mathbf{Q})_{i,j}(\mathbf{P}^T\mathbf{T}\mathbf{Q})_{k,l}\ell_A(\hat{A}_{i,k}, A_{j,l})h_jh_l.
\end{aligned}$$

Denoting $\tilde{\mathbf{T}} = \mathbf{P}^T\mathbf{T}\mathbf{Q}$ we have that

$$\begin{aligned}
\text{PMFGW}(\hat{y}', \mathcal{P}(g')) &= \min_{\tilde{\mathbf{T}} \in \pi_M} \sum_{i,j} \tilde{T}_{i,j}\ell_h(\hat{h}_i, h_j) + \sum_{i,j} \tilde{T}_{i,j}\ell_f(\hat{\mathbf{f}}_i, \mathbf{f}_j)h_j \\
&\quad + \sum_{i,j,k,l} \tilde{T}_{i,j}\tilde{T}_{k,l}\ell_A(\hat{A}_{i,k}, A_{j,l})h_jh_l \\
&= \text{PMFGW}(\hat{y}, \mathcal{P}(g)).
\end{aligned}$$

$\square$

We now provide a more detailed version of Proposition 3

**Definition 1.** *We say that $\ell : \hat{\mathcal{X}} \times \mathcal{X} \mapsto \mathbb{R}$ is positive when for any $x, y \in \hat{\mathcal{X}} \times \mathcal{X}$, $\ell(x, y) \geq 0$ with equality if and only if $x = y$.*

**Proposition 7** (Positivity). *Let us assume that $\ell_h : [0,1] \times \{0,1\} \mapsto \mathbb{R}, \ell_f : \mathbb{R}^d \times \mathbb{R}^d \mapsto \mathbb{R}$ and $\ell_A : [0,1] \times \{0,1\} \mapsto \mathbb{R}$ are positive. Then we have that for any $\hat{y} \in \hat{\mathcal{Y}}, g \in \mathcal{G}$ :*

- *i) $PMFGW(\hat{y}, \mathcal{P}(g)) \geq 0$*

- *ii) There is equality if and only if $\hat{y} \sim \mathcal{P}(g)$*

- *iii) In that case $\mathcal{P}^{-1}\mathcal{T}(\hat{y}) \sim g$*

*Proof.* The direct implication of ii) is the only statement that is not trivial. First, let us show that if PMFGW$(\hat{y}, \mathcal{P}(g)) = 0$, the optimal transport $\mathbf{T}^*$ is a permutation. Recall that any transport plan is

a convex combination of permutations [6] i.e. there exist $\lambda_1, ..., \lambda_K \in ]0, 1]$ and $\mathbf{P}_1, ..., \mathbf{P}_K \in \sigma_M$ such that $\sum_{k=1}^{K} \lambda_k = 1$ and $\mathbf{T}^* = \sum_{k=1}^{K} \lambda_k \mathbf{P}_k$. Thus

$$0 = \langle \mathbf{T}^*, \mathbf{C} \rangle + \langle \mathbf{T}^*, \mathbf{L} \otimes \mathbf{T}^* \rangle \tag{8}$$

$$= \sum_{k=1}^{K} \lambda_k \langle \mathbf{P}_k, \mathbf{C} \rangle + \sum_{k=1}^{K} \lambda_k^2 \langle \mathbf{P}_k, \mathbf{L} \otimes \mathbf{P}_k \rangle + \sum_{k \neq l}^{K} \lambda_k \lambda_l \langle \mathbf{P}_k, \mathbf{L} \otimes \mathbf{P}_l \rangle. \tag{9}$$

This is a sum of positive terms, thus all terms are null and in particular, for any $k$

$$0 = \langle \mathbf{P}_k, \mathbf{C} \rangle + \langle \mathbf{P}_k, \mathbf{L} \otimes \mathbf{P}_k \rangle. \tag{10}$$

Thus all the $\mathbf{P}_k$ are optimal transport plans. In the following, we chose one of them and denote it $\mathbf{P}$. Moving back to the developed formulation of PMFGW we get that

$$0 = \text{PMFGW}(\hat{y}, \mathcal{P}(g)) = \sum_{i,j} P_{i,j} \ell_h(\hat{h}_i, h_j) + \sum_{i,j} P_{i,j} \ell_f(\hat{\mathbf{f}}_i, \mathbf{f}_j) h_j + \sum_{i,j,k,l} P_{i,j} P_{k,l} \ell_A(\hat{A}_{i,k}, A_{j,l}) h_j h_l.$$

Once again this is a sum of positive terms thus for all $i, j, k$, and $l$

$$0 = P_{i,j} \ell_h(\hat{h}_i, h_j) = P_{i,j} \ell_f(\hat{\mathbf{f}}_i, \mathbf{f}_j) h_j = P_{i,j} P_{k,l} \ell_A(\hat{A}_{i,k}, A_{j,l}) h_j h_l$$

and thus

$$0 = \ell_h((\mathbf{P}^T \hat{\mathbf{h}})_j, h_j) = \ell_f((\mathbf{P}^T \hat{\mathbf{F}})_j, F_j) h_j = \ell_A((\mathbf{P}^T \hat{\mathbf{A}} \mathbf{P})_{j,l}, A_{j,l}) h_j h_l.$$

And from the positivity of $\ell_h, \ell_f$ and $\ell_A$ we get that: $\mathbf{P}^T \hat{\mathbf{h}} = \mathbf{h}$, $\mathbf{P}^T \hat{\mathbf{F}}[: m] = \mathbf{F}[: m]$ and $\mathbf{P}^T \hat{\mathbf{A}} \mathbf{P}[: m, : m] = \mathbf{A}[: m, : m]$. Since the nodes $i > m$ are not activated, by abuse of notation we simply write $\mathbf{P}^T \hat{\mathbf{F}} = \mathbf{F}$ and $\mathbf{P}^T \hat{\mathbf{A}} \mathbf{P} = \mathbf{A}$. This concludes that $\hat{y} \sim \mathcal{P}(g)$.

□

### B.3 PMFGW and Partial Fused Gromov Wasserstein

Following Chapel et al. [13], we define an OT relaxation of the Partial Matching problem.

For a large graph $\hat{g} = (\hat{\mathbf{F}}, \hat{\mathbf{A}}) \in \mathcal{G}_M$ and a smaller graph $g = (\mathbf{F}, \mathbf{A}) \in \mathcal{G}_m$, the set of transport plan transporting a subgraph of $\hat{g}$ to $g$ can be defined as

$$\pi_{M,m} = \{ \mathbf{T} \in [0, 1]^{M \times m} \mid \mathbf{T} \mathbb{1}_m \leq \mathbb{1}_M, \mathbf{T}^T \mathbb{1}_M = \mathbb{1}_m, \mathbb{1}_M^T \mathbf{T} \mathbb{1}_m = m \}$$

and the associated partial Fused Gromov Wasserstein distance is

$$\text{partialFGW}(\hat{g}, g) = \min_{\mathbf{T} \in \pi_{M,m}} \sum_{i=1}^{M} \sum_{j=1}^{m} \mathbf{T}_{i,j} \ell_F(\hat{\mathbf{f}}_i, \mathbf{f}_j) + \sum_{i,k=1}^{M} \sum_{j,l=1}^{m} \mathbf{T}_{i,j} \mathbf{T}_{k,l} \ell_A(A_{i,k}, A_{j,l}).$$

In the following, we show that $\text{partialFGW}(\hat{g}, g)$ is equivalent to the padded Fused Gromov Wasserstein distance defined as

$$\text{paddedFGW}(\hat{g}, g) = \min_{\mathbf{T} \in \pi_M} \sum_{i=1}^{M} \sum_{j=1}^{m} \mathbf{T}_{i,j} \ell_F(\hat{\mathbf{f}}_i, \mathbf{f}_j) + \sum_{i,k=1}^{M} \sum_{j,l=1}^{m} \mathbf{T}_{i,j} \mathbf{T}_{k,l} \ell_A(A_{i,k}, A_{j,l}).$$

*Lemma* 5. Any transport plan $\mathbf{T} \in \pi_M$ has the form $\mathbf{T} = (\mathbf{T}_p \quad \mathbf{T}_2)$ where $\mathbf{T}_p$ is a partial transport plan i.e. $\mathbf{T}_p \in \pi_{M,m}$.

*Proof.* Let us check that $\mathbf{T}_p$ is in $\pi_M$.

- $\mathbb{1}_M = \mathbf{T}\mathbb{1}_M = \mathbf{T}_p\mathbb{1}_m + \mathbf{T}_2\mathbb{1}_{M-m} \geq \mathbf{T}_p\mathbb{1}_m$

- $\mathbb{1}_M = \mathbf{T}^T\mathbb{1}_M = \begin{pmatrix} \mathbf{T}_p^T\mathbb{1}_M \\ \mathbf{T}_2^T\mathbb{1}_M \end{pmatrix}$. And thus $\mathbf{T}_p^T\mathbb{1}_M = \mathbb{1}_m$

- From the previous we immediately get that $\mathbb{1}_M^T\mathbf{T}_p\mathbb{1}_m = m$

$\square$

*Lemma* 6. For any partial transport plan $\mathbf{T}_p \in \pi_{M,m}$ there exist $T_2 \in \mathbb{R}^{M \times (M-m)}$ such that $\mathbf{T} = (\mathbf{T}_p \quad \mathbf{T}_2) \in \pi_M$.

*Proof.* Let us define $p = \mathbb{1}_M - \mathbf{T}_p\mathbb{1}_m$. This is the mass of the larger graph that is not matched by $\mathbf{T}_p$. Note that since $\mathbf{T}_p \in \pi_{M,m}$ we have that $p \geq 0$. Thus we can set $\mathbf{T}_2 = \frac{1}{M-m}p\mathbb{1}_{M-m}^T$ i.e. we spread the remaining mass uniformly across the padding nodes. Let us check that $\mathbf{T} = (\mathbf{T}_p \quad \mathbf{T}_2) \in \pi_M$ is indeed a valid transport plan.

- $\mathbf{T}\mathbb{1}_M = \mathbf{T}_p\mathbb{1}_m + \mathbf{T}_2\mathbb{1}_{M-m} = \mathbf{T}_p\mathbb{1}_m + p = \mathbb{1}_m$

- $\mathbf{T}^T\mathbb{1}_M \quad = \quad \begin{pmatrix} \mathbf{T}_p^T\mathbb{1}_M \\ \mathbf{T}_2^T\mathbb{1}_M \end{pmatrix} \quad = \quad \begin{pmatrix} \mathbb{1}_m \\ \frac{1}{M-m}(p^T\mathbb{1}_M)\mathbb{1}_{M-m} \end{pmatrix} \quad = \quad \begin{pmatrix} \mathbb{1}_m \\ \frac{1}{M-m}(\mathbb{1}_M^T\mathbb{1}_M - \mathbb{1}_m^T\mathbf{T}_p^T\mathbb{1}_M)\mathbb{1}_{M-m} \end{pmatrix} = \begin{pmatrix} \mathbb{1}_m \\ \mathbb{1}_{M-m} \end{pmatrix} = \mathbb{1}_M$

$\square$

**Proposition 8.** *paddedFGW and partialFGW are equal and any optimal plan $\mathbf{T}^*$ of paddedFGW has the form $\mathbf{T}^* = (T_p^*, T_2)$ where $T_p^*$ is optimal for partialFGW.*

*Proof.* Follows directly from the two previous lemmas. $\square$

*Remark* 7. Since paddedFGW is equivalent to the proposed PMFGW loss if and only if $\ell_h$ is set to a constant, proposition 4 is a direct corollary of Proposition 8.

*Remark* 8. The algorithm proposed to compute PMFGW can be applied to paddedFGW and thus to partialFGW. Hence, we have indirectly introduced an alternative to the algorithm of Chapel et al. [13]. Further comparisons are left for future work.

## C   Encoding Any Input To Graph

**Philosophy of the Any2Graph encoder**   Any2Graph is compatible with different types of inputs, given that one selects the appropriate encoder. The role of the encoder is to extract a set of feature vectors from the inputs $x$ i.e. each input is mapped to a list of $k$ feature vectors of dimension $d_e$ where $k$ is not necessarily fixed. This is critically different from extracting a unique feature vector ($k = 1$). If $k$ is set to 1, the rest of the architecture must reconstruct an entire graph from a single vector, and the architecture is akin to that of an auto-encoder. In Any2Graph, we avoid this undesirable bottleneck by opting for a richer ($k > 1$) and more flexible ($k$ is not fixed) representation of the input. The $k$ feature vectors are then fed to a transformer which is well suited to process sets of different sizes. Since the transformer module is permutation-invariant any meaningful ordering is lost in the process. To alleviate this issue, we add positional encoding to the feature vectors whenever the ordering carries information. Finally, note that the encoder might highly benefit from pre-training whenever applicable; but this goes beyond the scope of this paper.

We now provide a general description of the encoders that can be used for each input modality.

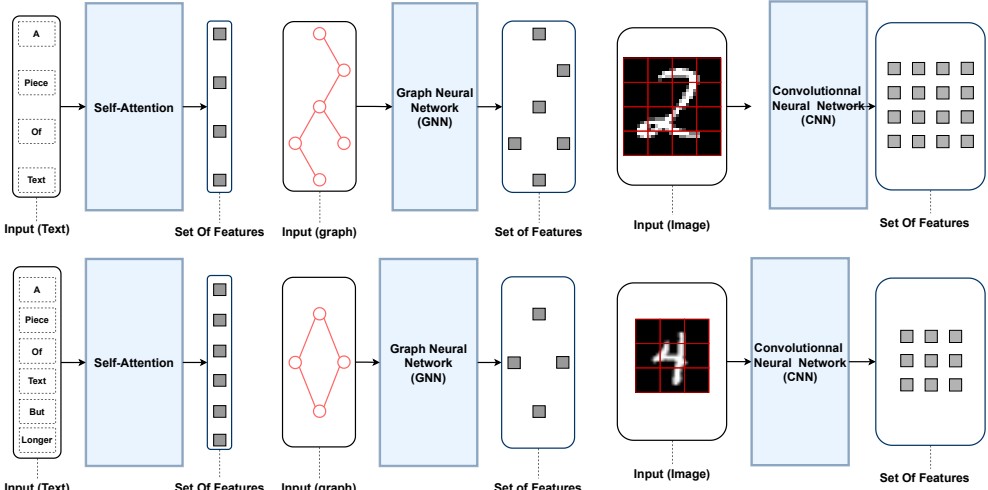

Figure 7: Illustration of encoders extracting $k$ features vectors for different input modalities. For text/fingerprint, $k$ is the number of input tokens. For graphs, $k$ is the size of the input graph. For images, $k$ depends on the resolution of the image and the CNN kernel size.

**Images** For Image2Graph task we use Convolutional Neural Networks (CNN) as suggested in [36]. From an input image of shape $h \times w \times c$ the CNN outputs a tensor of shape $H \times W \times C$ which is seen as $H \times W$ feature vectors of dimension $C$. The raw output of the CNN is reshaped and passed through a linear layer to produce the final output of shape $H \times W \times d_e$. Since the ordering of the $H \times W$ features carries spatial information we add positional encoding accordingly.

**Fingerprint/text** For tasks where the input is a list of tokens (e.g. Text2Graph or Fingerprint2Graph) we use the classical NLP pipeline: each token is transformed into a vector by an embedding layer and the list of vectors is then processed by a transformer encoder module. In text2graph the tokens ordering carries semantic meaning and positionnal encoding should be added. On the contrary, in Fingerprint2Graph, the fingerprint ordering carries no information and the permutation invariance of the transformer module is a welcomed property.

**Graph** For a graph2graph task (not featured in this paper) we would suggest using a Graph Neural Network (GNN) [52]. A GNN naturally extracts $k$ feature vectors from an input graph, where $k$ is the number of nodes in the input graph. No positional encoding is required.

**Vector** We explore a Vect2Graph task in Appendix D. The naive encoder we use is composed of $k$ parallel MLPs devoted to the extraction of the $k$ feature vectors. This approach is arguably simplistic and more suited encoders should be considered depending on the type of data.

# D COLORING: a new synthetic dataset for benchmarking Supervised Graph Prediction

We introduce *Coloring*, a new synthetic dataset well suited for benchmarking SGP methods. The main advantages of *Coloring* are:

- The output graph is uniquely defined from the input image.
- The complexity of the task can be finely controlled by picking the distribution of the graph sizes, the number of node labels (colors) and the resolution of the input image.
- One can generate as many pairs (inputs, output) as needed to explore different regimes, from abundant to scarce data.

To generate a new instance of *Coloring*, we apply the following steps:

- 0) Sample the number of nodes (graph size) $m$. In this paper, we sample uniformly on some interval $[M_{\min}, M_{\max}]$.

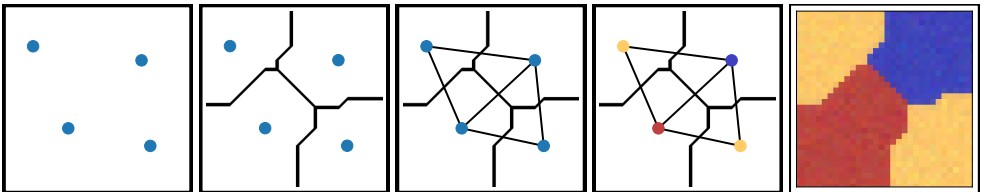

Figure 8: Illustration of the five steps to follow to generate a new instance of *Coloring*.

- 1) Sample $m$ centroids on $[0, 1] \times [0, 1]$. In this paper, we sample the centroids as uniform i.i.d. variables.
- 2) Partition $[0, 1] \times [0, 1]$ (the image) in a Voronoi diagram fashion [30]. In this paper, we use the $L_1$ distance. and an image of resolution $H \times H$.
- 3) Create the associated graph i.e. each node is a region of the image and two nodes are linked by an edge whenever the two associated regions are adjacent.
- 4) Color the graph with $K > 4$ colors. In this paper, we use $K = 4$. A coloring is said to be valid whenever no adjacent nodes have the same color. Note that graph coloring is known to be NP-complete [23].
- 5) Color the original image accordingly.

As highlighted above, *Coloring* is a flexible dataset. Beyond the default dataset simply referred as *Coloring* we also explore 2 variations in our experiments. *ColoringBig* is a more challenging dataset that features larger graphs. *ColoringVect* is a variation of *Coloring* where the input image is flattened and treated as a vector allowing us to explore a synthetic Vect2Graph task. The properties of these datasets, along with the performances of Any2Graph are reported in table 3. We hope that *Coloring* will be used to benchmark future SGP methods.

Table 3: Summary of the properties of the 3 variations of *Coloring* considered in this paper. We also report the test edit distance achieved by the different models. For FGWBary we report the best performing variant that is ILE for *ColoringVect* and NN for *Coloring*. None scales to *ColoringBig*.

| DATASET | $M_{\text{MIN}}$ | $M_{\text{MAX}}$ | H | NUMBER OF SAMPLES | ANY2GRAPH | RELATIONFORMER | FGWBARY-NN |
|---|---|---|---|---|---|---|---|
| *ColoringVect* | 4 | 6 | 16 | 100K | 0.46 | 1.40 | 2.09 |
| *Coloring* | 6 | 10 | 32 | 100K | 0.20 | 5.47 | 6.73 |
| *ColoringBig* | 10 | 15 | 64 | 200K | 1.01 | 8.91 | N.A. |

# E  Additional Details On The Experimental setting

## E.1  Datasets

In this paper, we consider five datasets for which we provide a variety of statistics in Table 4. *Coloring* is a new synthetic dataset which we describe in detail in Appendix D. *Toulouse* (resp. *USCities*) is a Sat2Graph dataset [5] where the inputs are images of size $64 \times 64$ (resp. $128 \times 128$). *QM9* [50] and *GDB13* [7] are datasets of small molecules which we use to address the Fingerprint2Graph task. Here, we compute a fingerprint representation of the molecule and attempt to reconstruct the original molecule from this loss representation. Following Ucak et al. [42] we use the Morgan Radius-2 fingerprint [26] which represents a molecule by a bit vector of size $2048$, where each bit represents the presence/absence of a given substructure. Finally, we feed our model with the list of non-zeros bits, i.e. the list of substructures (tokens) present in the molecule. The list of substructures has a min/average/max length of $2/21/27$ for *QM9* and $7/29/36$ for *GDB13*.

Table 4: Table summarizing the properties of the datasets considered.

| DATASET | SIZE (TRAIN/TEST/VALID) | NODES (MIN/MEAN/MAX) | EDGES (MIN/MEAN/MAX) | INPUT MODALITY | NODE FEATURES |
|---|---|---|---|---|---|
| *Coloring* | 100K/10K/10K | 4/7.0/10 | 3/10.9/22 | RGB IMAGES | 4 CLASSES (COLORS) |
| *Toulouse* | 80K/10K/10K | 3/6.1/9 | 2/5.0/14 | GREY IMAGES | 2D POSITIONS |
| *USCities* | 130K/10K/10K | 2/7.5/17 | 1/5.8/20 | GREY IMAGES | 2D POSITIONS |
| *QM9* | 120K/10K/10K | 1/8.8/9 | 0/9.4/13 | LIST OF TOKENS | 4 CLASSES (ATOMS) |
| *GDB13* | 1300K/70K/70K | 5/12.7/13 | 5/15.15/18 | LIST OF TOKENS | 5 CLASSES (ATOMS) |

## E.2  Training And Architecture Hyperparameters

**Encoder**  We follow the guidelines established in C for the choice of the encoder. In particular, all encoders for *Coloring*, *Toulouse* and *USCities* are CNNs. The encoder for *Coloring* is a variation of Resnet18 [22], where we remove the first max-pooling layer and the last two blocks to accommodate for the low resolution of our input image. We proceed similarly for *Toulouse* except that we only remove the last block. For *USCities* we keep the full Resnet18. For the Fingerprint2Graph datasets, we use a transformer encoder. In practice, this transformer encoder and that of the encoder-decoder module are merged to avoid redundancy. All encoders end with a linear layer projecting the feature vectors to the hidden dimension $d_e$.

**Transformer**  We use the Pre-LN variant Xiong et al. [51] of the transformer encoder-decoder model as described in [43]. To reduce the number of hyperparameters, encoder and decoder modules both consist of stacks of $N_\tau$ layers, with $N_h$ heads and the hidden dimensions of all MLP is set to $4 \times d_e$.

**Decoder**  All MLPs in the decoder module have one hidden layer.

**Optimizer**  We train all neural networks with the Adam optimizer Kingma and Ba [24], learning rate $\eta$, 8000 warm-up steps and all other hyperparameters set to default values. We also use gradient clipping with a max norm set to 0.1.

All hyperparameters are given in Table 5.

Table 5: Hyperparameters used to train our models. We also report the total training time on a NVIDIA V100.

| DATASET | PMFGW | | | ARCHITECTURE | | | | | OPTIMIZATION | | |
|---|---|---|---|---|---|---|---|---|---|---|---|
| | $\alpha_H$ | $\alpha_F$ | $\alpha_A$ | $d_e$ | $N_\tau$ | $N_h$ | $M$ | $\eta$ | BATCHSIZE | STEPS | TIME |
| *Coloring* | 1 | 1 | 1 | 256 | 3 | 8 | 12 | 3E-4 | 128 | 75K | 4H |
| *Toulouse* | 1 | 5 | 1 | 256 | 4 | 8 | 12 | 1E-4 | 128 | 100K | 8H |
| *USCities* | 2 | 5 | 0.5 | 256 | 4 | 8 | 20 | 1E-4 | 128 | 150K | 14H |
| *QM9* | 1 | 1 | 1 | 128 | 3 | 4 | 12 | 3E-4 | 128 | 150K | 6H |
| *GDB13* | 1 | 1 | 1 | 512 | 5 | 8 | 15 | 3E-4 | 256 | 150K | 24H |

### E.3 Metrics

In the following, we provide a detailed description of the metrics reported in table 1.

**Graph Level** First we report the PMFGW loss between continuous prediction $\hat{y}$ and padded target $\mathcal{P}(g)$. For this computation, we set $\alpha$ to the values displayed in table 5.

$$\text{PMFGW}(\hat{y}, \mathcal{P}(g))$$

Then we report the graph edit distance Gao et al. [19] between predicted graph $\mathcal{P}^{-1}\mathcal{T}\hat{y}$ and target $g$ which we compute using Pygmtools Wang et al. [48]. All edit costs (nodes and edges) are set to 1. Note that for *Toulouse* and *USCities*, node labels are 2D positions and we consider two nodes features to be equal (edit cost of 0) whenever the L2 distance is smaller than 5% than the image width.

$$\text{EDIT}(\mathcal{P}^{-1}\mathcal{T}\hat{y}, g)$$

Finally, we report the Graph Isomorphism Accuracy GI ACC that is

$$\text{GI ACC}(\hat{y}, g) = \mathbb{1}[\text{EDIT}(\mathcal{P}^{-1}\mathcal{T}\hat{y}, g) = 0]$$

**Node level** Recall that for a prediction $\hat{y} = (\hat{\mathbf{h}}, \hat{\mathbf{F}}, \hat{\mathbf{A}})$ the size of the predicted graph is $\hat{m} = ||\hat{h} > 0.5||_1$. Denoting $m$ the size of the target graph we report the size accuracy:

$$\text{SIZE ACC}(\hat{y}, g) = \mathbb{1}[\hat{m} = m].$$

The remaining node and edge-level metrics need the graphs to have the same number of nodes. To this end, we select the $m$ nodes with the highest probability $\hat{h}_i$, resulting in a graph $\hat{g} = (\tilde{\mathbf{F}}, \tilde{\mathbf{A}})$ with ground truth size. This is equivalent to assuming that the size of the graph is well predicted. Then we use Pygmtools to compute a one-to-one matching $\sigma$ between the nodes of $\hat{g}$ and $g$ that can be used to align graphs (we use the matching that minimizes the edit distance). In the following, we assume that $g$ and $\hat{g}$ have been aligned. We can now define the node accuracy NODE ACC as

$$\text{NODE ACC}(\hat{g}, g) = \frac{1}{m} \sum_{i=1}^{m} \mathbb{1}[\tilde{F}_i = F_i],$$

which is the average number of node features that are well predicted.

**Edge level** Since the target adjacency matrices are typically sparse, the edge prediction accuracy is a poorly informative metric. To mitigate this issue we report both Edge Precision and Edge Recall :

$$\text{EDGE PREC.}(\hat{g}, g) = \frac{\sum_{i,j=1}^{m} \mathbb{1}[\tilde{A}_{i,j} = 1, A_{i,j} = 1]}{\sum_{i,j=1}^{m} \mathbb{1}[\tilde{A}_{i,j} = 1]}$$

$$\text{EDGE PREC.}(\hat{g}, g) = \frac{\sum_{i,j=1}^{m} \mathbb{1}[\tilde{A}_{i,j} = 1, A_{i,j} = 1]}{\sum_{i,j=1}^{m} \mathbb{1}[A_{i,j} = 1]}$$

All those metrics are then averaged other the test set.

### E.4 Compute resources

We estimate the total computational cost of this work to be approximately 1000 hours of GPU (mostly Nvidia V100). We estimate that up to 70% of this computation time was used in preliminary work and experiments that did not make it to the paper.

## F  Additional Experiments and figures

### F.1  Learning dynamic

The PMFGW loss is composed of three terms, two of them are linear and account for the prediction of the nodes and their features, one is quadratic and accounts for the prediction of edges. The last term is arguably the harder to minimize for the model, as a consequence, we observe that the training performs best when the two first terms are minimized first which then guides the minimization of the structure term. In other words, the model must first learn to predict the nodes before addressing their relationship. Fortunately, this behavior naturally arises in Any2Graph as long as $\alpha_A$, the hyperparameter controlling the importance of the quadratic term, is not too large. This is illustrated in figure 9.

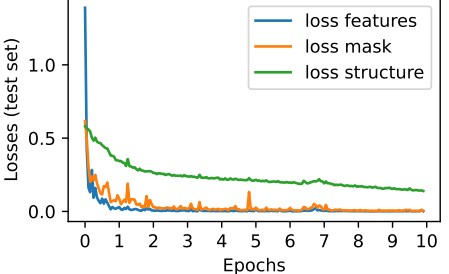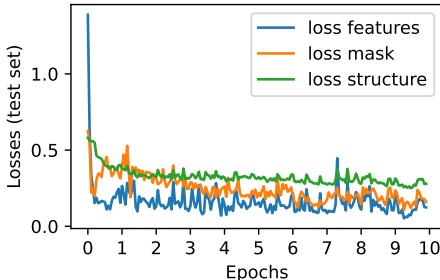

Figure 9: First epochs of training for *Coloring*. The test values of the 3 components of the loss are reported. On the left (resp. right) $\alpha$ is set to $[1, 1, 1]$ (resp. $[1,1,10]$). In the first scenario, the first two terms of the loss are learned very fast and the structure is optimized next. In the second scenario, setting $\alpha_A = 10$ prevents this desirable learning dynamic.

For the datasets where many nodes in the graphs share the same features (*QM9* and *GDB13*) the good prediction of the nodes and their features is not enough to guide the prediction of the edges and this desirable dynamic does not occur. This motivates us to perform Feature Diffusion (FD) before training. The diffused node features carry a portion of the structural information. This makes the node feature term slightly harder to minimize but in turn, the subsequent prediction of the structure is much easier and we recover the previous dynamic. This is illustrated in figure 10.

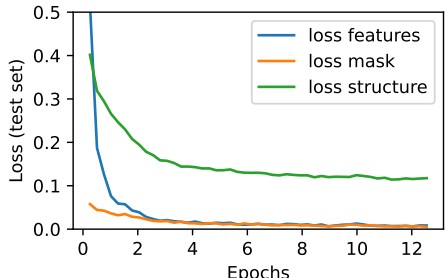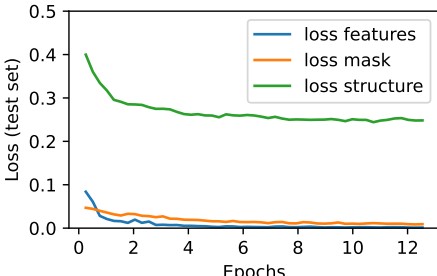

Figure 10: First epochs of training for *GDB13*. The test values of the 3 components of the loss are reported. On the left, we perform FD before training, on the right, we leave node features unchanged. We observe that the feature loss decreases slightly slower with FD (the features are more complex) but the minimization of the structure term is largely accelerated.

### F.2  Effect of the OT relaxation on the performances

As stated in 2, we adopted the OT point of view when designing Any2Graph. In practice, this means that we do not project the OT plan back to the set of permutations with a Hungarian matcher before plugging it in the loss as in Simonovsky and Komodakis [37]. Testing the effect of adding this extra

step we observed a 5% to 10% increase of the edit distance across datasets (table 6) along with a more unstable training curve (figure 11). This confirms that a continuous transport plan provides a slightly more stable gradient than a discrete permutation, which aligns with the findings of De Plaen et al. [16] on the similar topic of object detection.

| Dataset | Coloring | Toulouse | USCities | QM9 | GDB13 |
|---|---|---|---|---|---|
| ED without Hungarian | **0.20** | **0.13** | **1.86** | 2.13 | **3.63** |
| ED with Hungarian | 0.23 | 0.15 | 2.03 | **2.08** | 3.86 |

Table 6: Effect of adding Hungarian Matching on the performances evaluated with the test edit distance. We observe, that Hungarian Matching slightly decreases the performances on all datasets but *QM9*.

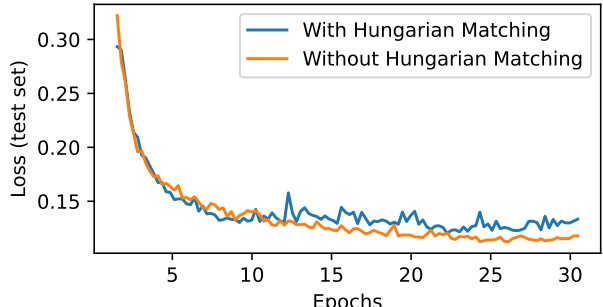

Figure 11: First epochs of training for *GDB13* with and without projection of the optimal transport plan to the set of permutations with Hungarian matching. Hungarian matching slightly decreases the performances and induces more oscillations of the loss, which could be explained by a less stable gradient.

## F.3 Repeatability

We checked the robustness of Any2Graph to the seed used for training. For each dataset, we ran 5 times the whole training pipeline with 5 random seeds. The results are displayed in table 7. Notably, we observe that all state-of-the-art performances observed in table 1 hold even if we had reported the worst seed (we reported the first seed).

| Dataset | Coloring | Toulouse | USCities | QM9 | GDB13 |
|---|---|---|---|---|---|
| Edit. (Max) | 0.23 | 0.16 | 2.08 | 2.19 | 3.63 |
| Edit. (Mean) | 0.21 | 0.14 | 1.89 | 2.10 | 3.49 |
| Edit. (Min) | 0.2 | 0.13 | 1.7 | 2.03 | 3.43 |
| Edit. (Std) | 0.01 | 0.08 | 0.14 | 0.05 | 0.07 |

Table 7: For each dataset, we report the maximum, average, minimum and the standard deviation of the Any2Graph test edit distance over 5 random seeds.

# G Additional Qualitative Results

## G.1 Qualitative results on COLORING

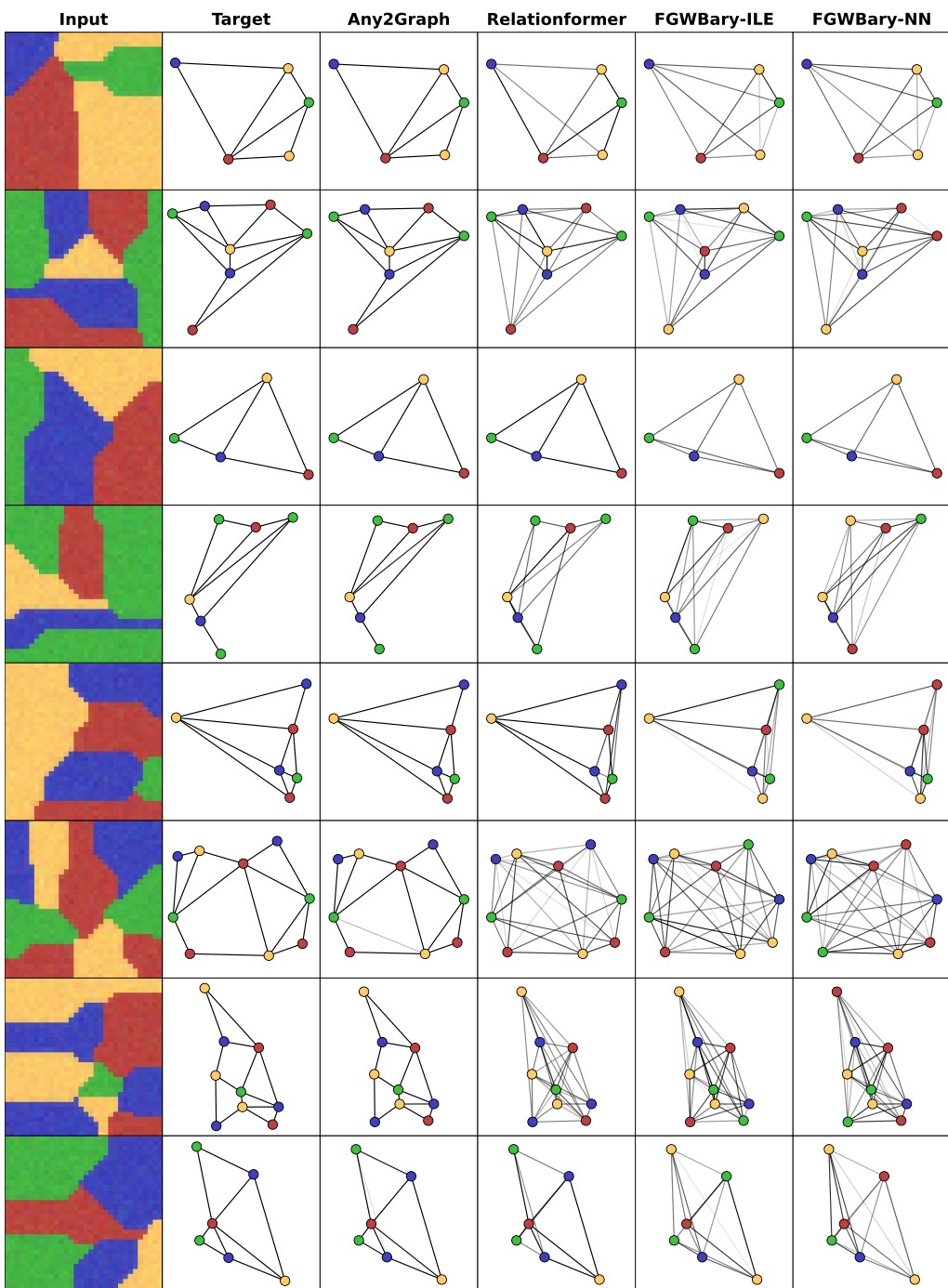

Figure 12: Graph prediction on the *Coloring* dataset.

## G.2 Qualitative results on QM9

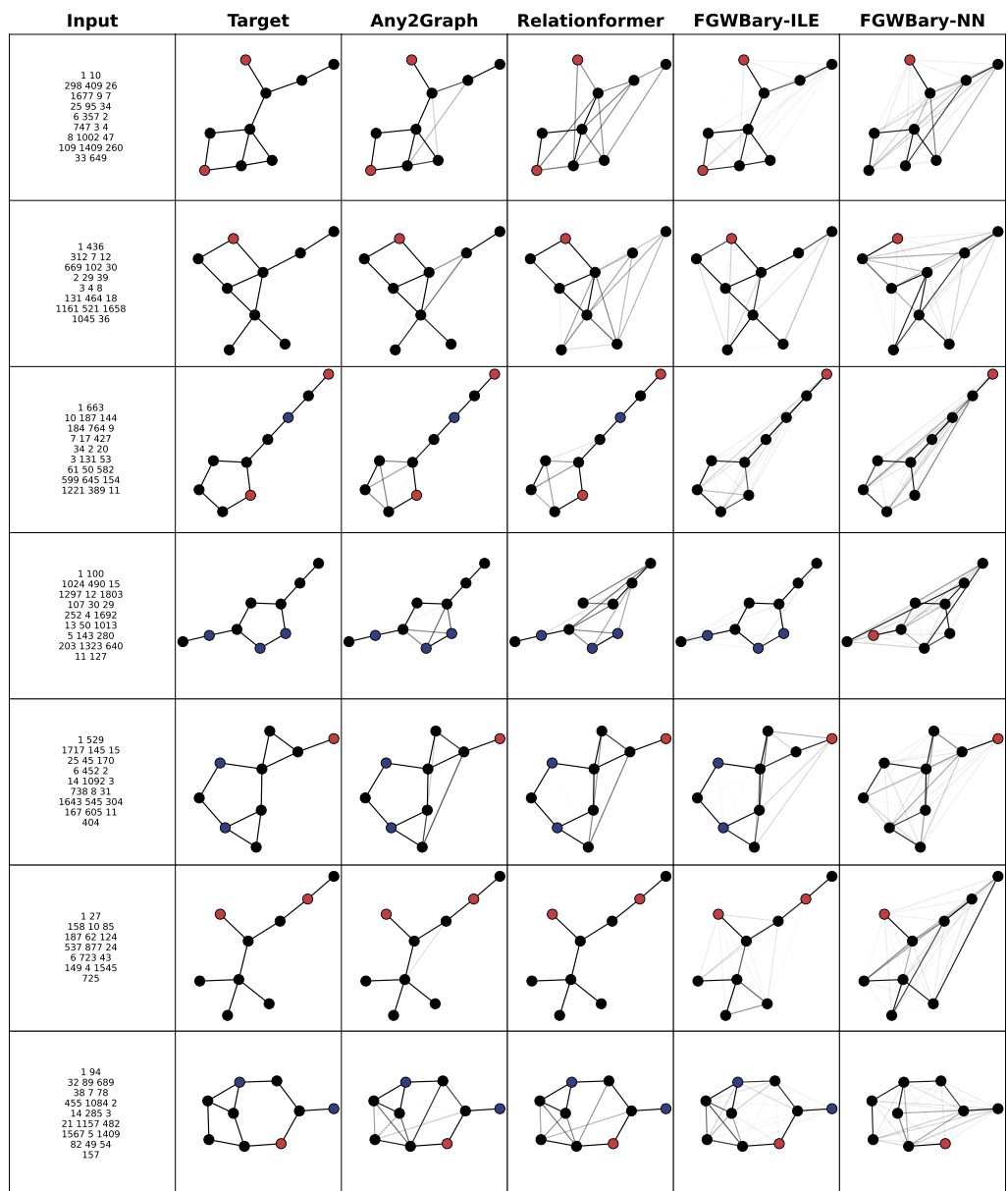

Figure 13: Graph prediction on the *QM9* dataset.

## G.3   Qualitative results on GDB13

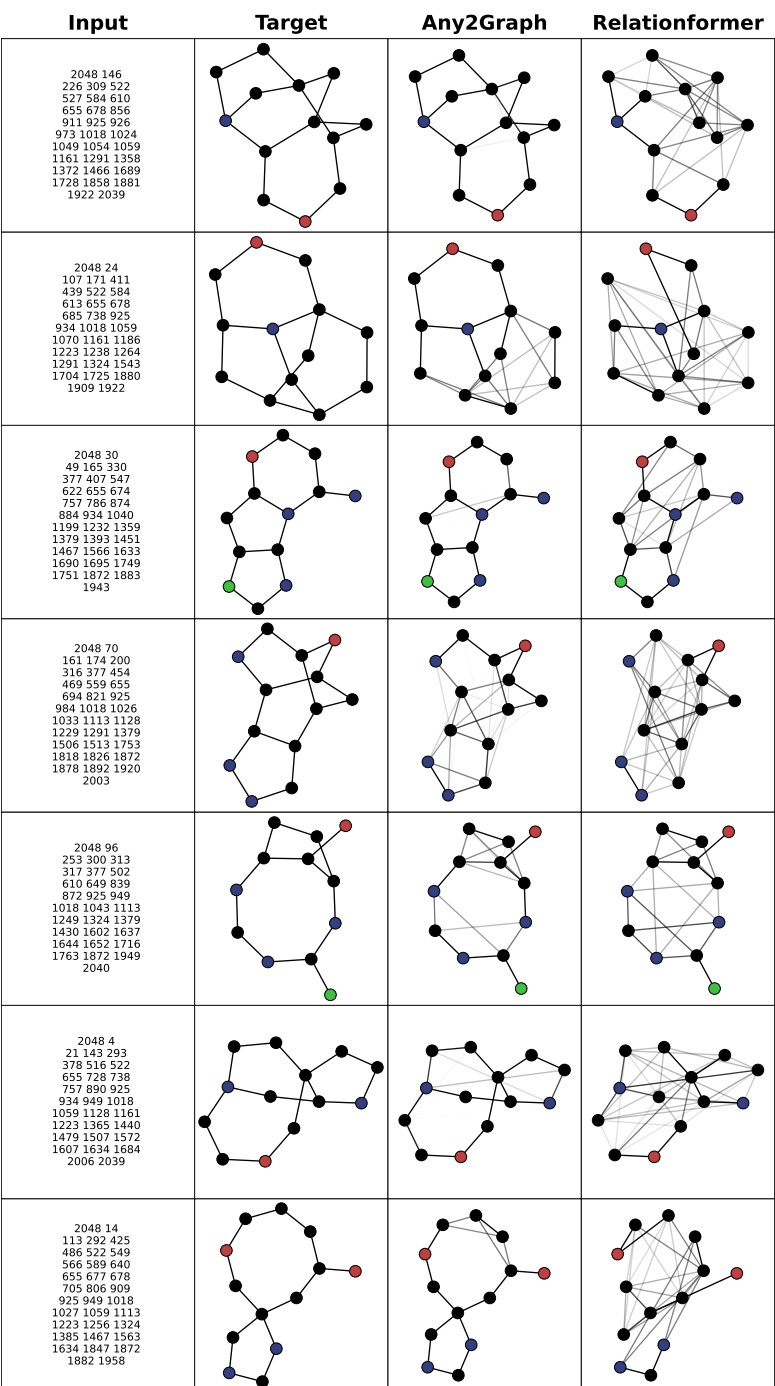

Figure 14: Graph prediction on the *GDB13* dataset.

## G.4 Qualitative results on TOULOUSE

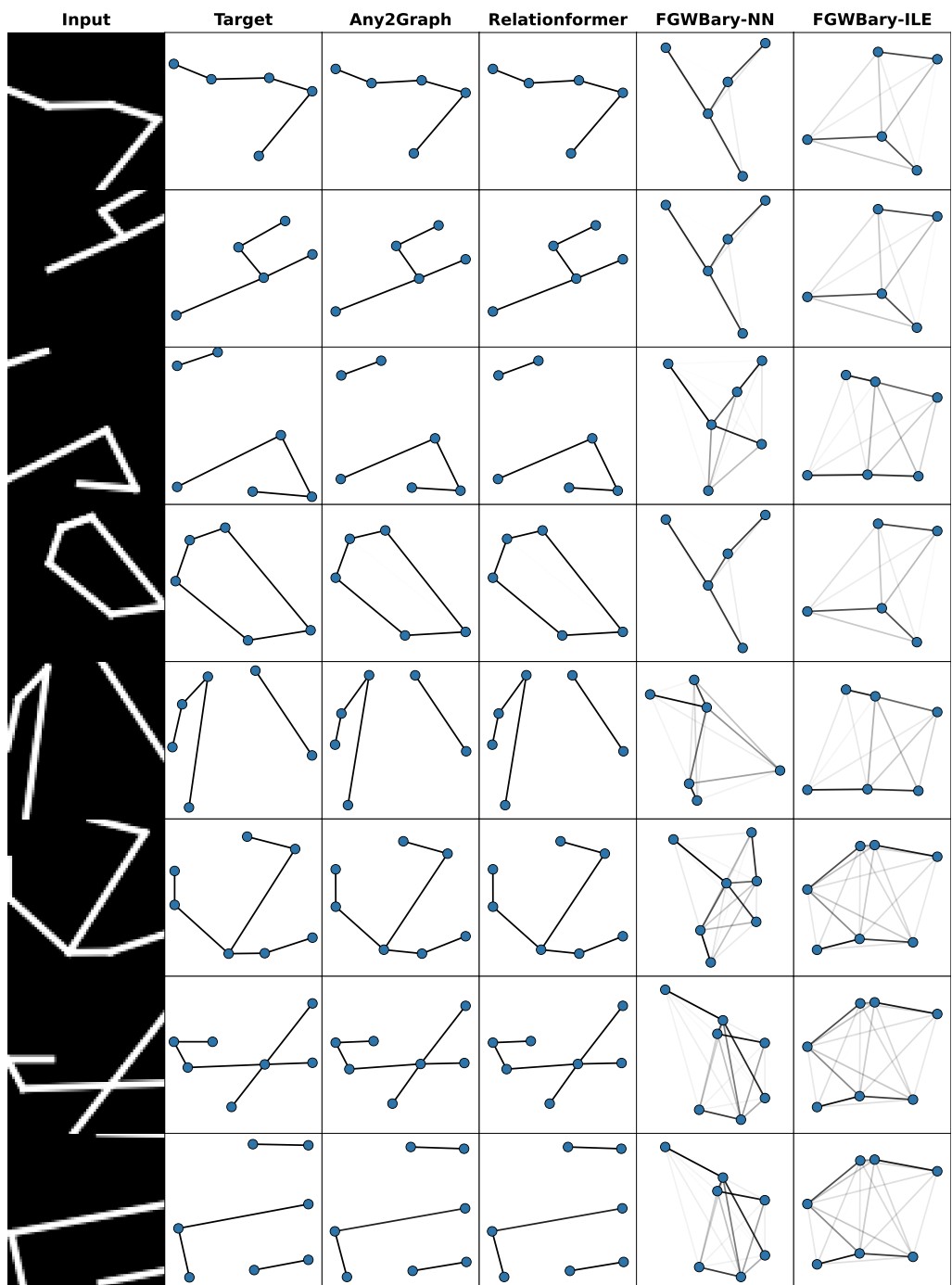

Figure 15: Graph prediction on the *Toulouse* dataset.

## G.5 Qualitative results on USCities

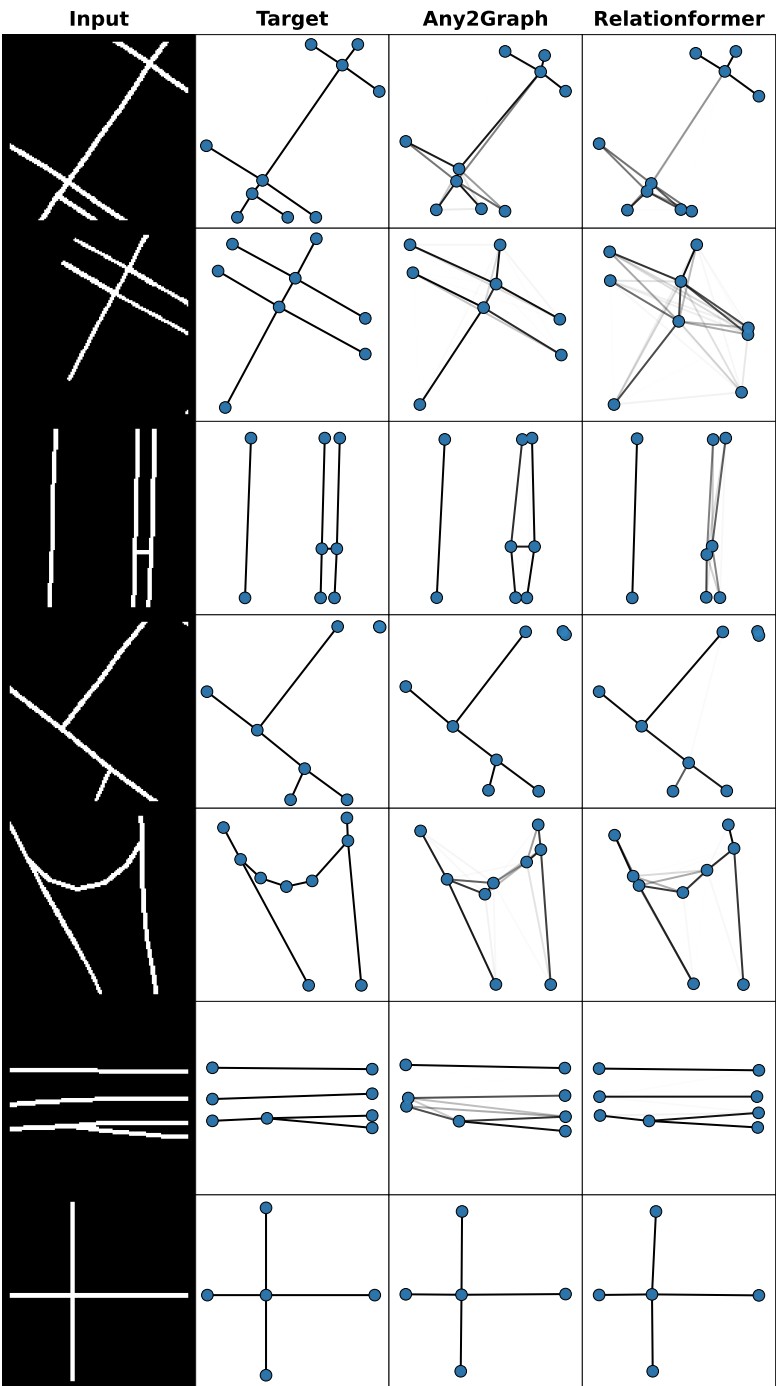

Figure 16: Graph prediction on the *USCities* dataset.

### G.6 Out of distribution performances

We tested if, once trained on Toulouse dataset, the predictive model is able to cope with out-of-distribution data. Figure 17 shows that this is the case on these toy images, that are not related to satellite images or road maps. We leave for future work the investigation of this property.

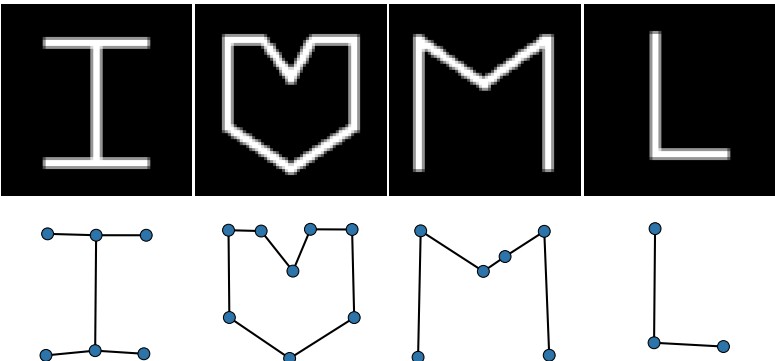

Figure 17: Any2Graph trained on Toulouse performing on out-of-distribution inputs. Input images are displayed on top row and prediction in the bottom row.

