# OpenReview forum: "Any2Graph: Deep End-To-End Supervised Graph Prediction With An Optimal Transport Loss"
_NeurIPS.cc/2024/Conference — NeurIPS 2024 spotlight_

### Official Review · Reviewer_ct8H · 2024-07-11

**Soundness:** 4
**Presentation:** 4
**Contribution:** 3
**Rating:** 7
**Confidence:** 3

**Summary:**

This paper presents a generic framework for end-to-end supervised graph prediction. The proposed framework can take different types of data as input and learn to output graphs. The core of the framework is a novel loss function (PMFGW) that enables generalizability in different scenarios. In the end, the paper demonstrates the capability and superior performance of Any2Graph over existing solutions on synthetic datasets as well as real-world datasets.

**Strengths:**

S1: The paper tackles an interesting and important problem: supervised graph predictions, which can potentially benefit many applications.

S2: The paper is very well-written and easy to follow. It effectively motivates the key challenge of the problem and clearly articulates the proposed idea step by step.

S3: The paper's primary contribution, PMFGW, is well-formulated. It also shows good performance across various metrics compared to prior solutions.

S4: The experiments in this paper are comprehensive and solid. They not only showcase PMFGW's high performance but also provide insights into its properties.

**Weaknesses:**

W1: The computational performance results could be more comprehensive. While the asymptotic time complexity is provided, it would be valuable to understand how training time scales with an increase in M. This information would help readers assess the practicality of using this approach for their specific use cases.

W2: It would be interesting to understand the failure cases produced by Any2Graph.

W3: Any2Graph produces continuous outputs; it would be interesting to know their distribution. How will the predicted graph affect the predicted graph if we use a different threshold, e.g., 0.1 or 0.9?

**Questions:**

Please check the weaknesses section.

**Limitations:**

The paper's contributions are unlikely to have any negative societal impacts.

---

> ### Author Rebuttal · Authors · 2024-08-06
>
> Thank you for reviewing our paper! All your questions are very interesting and answering them helped us to significantly improve the paper.
>
> **Weaknesses:**
>
> > W1: The computational performance results could be more comprehensive. While the asymptotic time complexity is provided, it would be valuable to understand how training time scales with an increase in M. This information would help readers assess the practicality of using this approach for their specific use cases.
>
> We report the training time for the different datasets in table 5 (appendix E.2) but we agree that the complexity analysis with respect to $M$ could be more detailed. Thank you for this important remark, we will fill this gap in the final version of the paper. In short: the cost of the transformer scales with $M^2$ and the cost of PMFGW scales with $kM^3$ where $k$ is the number of iteration required for the solver. We provide a novel plot for empirical estimation of $k$ in the pdf (figure 1).
>
> > W2: It would be interesting to understand the failure cases produced by Any2Graph.
>
> This is very important as well, thank you. We think that they are two types of failure cases. One the one hand, it can happen that the training dynamic fail to converge. We discuss this in detail in appendix F.1 along with simple methods to prevent this bad behaviour. On the other hand there is the case where $M$ is too large which we already discussed above.
>
> > W3: Any2Graph produces continuous outputs; it would be interesting to know their distribution. How will the predicted graph affect the predicted graph if we use a different threshold, e.g., 0.1 or 0.9?
>
> Once again this is an interesting point that is not discussed at all in the paper. We provide a few novel plots in the global response pdf. As you can see the model is very robust to the choice of the treshold (table 1 and 2)! This is because the model is quite confident in his prediction, as demonstrated in the histograms (figure 3 and 4).

---

### Official Review · Reviewer_bW5k · 2024-07-11

**Soundness:** 4
**Presentation:** 4
**Contribution:** 3
**Rating:** 8
**Confidence:** 3

**Summary:**

The authors propose a flexible framework for end-to-end Supervised Graph Prediction (SGP), called Any2Graph, capable of handling various types of input data. This framework leverages a novel, fully differentiable, and node permutation-invariant optimal transport-based loss called the Partially Masked Fused Gromov-Wasserstein (PMFGW) loss. Unlike the Fused Gromov-Wasserstein loss (FGW), which cannot compare graphs of different sizes, the PMFGW is size-agnostic. To satisfy this property, the discrete output space is relaxed to a continuous space. A discrete graph is mapped to a continuous graph using a padding operator. The PMFGW measures the discrepancy between the predicted continuous graph and the target padded graph. Essentially, PMFGW extends FGW by adding an additional term that ensures the padding of a node is well predicted, and the partial mapping of the second and third terms. The authors introduce three propositions demonstrating that PMFGW translates all properties of FGW to the new size-agnostic graph representation. The architecture of the end-to-end SGP framework is a modification of the Relationformer, ensuring versatility in terms of input data (not only images) and modifying the operation for computing the adjacency matrix. The authors showed that Any2Graph achieved state-of-the-art prediction performance at a very low computational inference cost on four real-world datasets characterized by different types of input data.

**Strengths:**

The paper is well-written, and both the description of the PMFGW loss and the architecture of the Any2Graph framework are clearly presented

The flexibility regarding the input data type and the size-agnostic graph representation properties provided by Any2Graph have the potential to significantly impact  the supervised graph prediction task, which play an important role in several applications from graphics to neuroscience.

The sound theoretical analysis proves that the novel PMFGW loss translates all the properties of FGW to the new size-agnostic representation based on the idea of mapping each graph to the corresponding continuous padded graph.

The numerical experiments are well-conducted with respect to:
- The comparison of state-of-the-art (SOTA) methods in terms of both accuracy and computational efficiency.
- The study of the robustness of Any2Graph concerning maximum graph size and the weight of the three terms of the PMFGW loss.

**Weaknesses:**

I think to improve the readability of of the paper I suggest to include a extend the description of the graph matching (section 2, paragraph 'Comparing graphs of the same size' by including a formal definition of the graph matching problem.

**Questions:**

At the end of section 2, the authors state that Unbalanced OT introduces several additional regularization parameters (I believe three) that are difficult to tune, especially in scenarios like SGP, where model predictions exhibit wide variability during training. While I agree, I was wondering if the authors have conducted any preliminary experiments using a UOT-based loss.

**Limitations:**

The authors have adequately addressed the limitations of the Any2Graph framework. Their work does not appear to have any potential negative societal impact.

---

> ### Author Rebuttal · Authors · 2024-08-06
>
> Thank you for your detailed review and positive feedback.
>
> > I think to improve the readability of of the paper I suggest to  extend the description of the graph matching (section 2, paragraph 'Comparing graphs of the same size')
>
> We will follow your suggestion to provide a more detailled introduction to the graph matching problem in the final version of the paper.
>
> > **(UOT as loss)** At the end of section 2, the authors state that Unbalanced OT introduces several additional regularization parameters (I believe three) that are difficult to tune, especially in scenarios like SGP, where model predictions exhibit wide variability during training. While I agree, I was wondering if the authors have conducted any preliminary experiments using a UOT-based loss.
>
> Thank you for this question! UOT is actually the first thing we tried and we are greatful for this opportunity to share the insight we gained from those initial attempts.
>
> To illustrate the discussion we can try to keep the exact same framework except we use FUGW [1] instead of PMFGW, that is we swap equation (5) with
>
> \begin{align}
>   \texttt{FUGW}(\hat{y},y) = \min_{\mathbf{T}\geq 0} \quad &\alpha_f  \sum_{i,j}T_{i,j}\ell_F(\hat{\mathbf{f}\}_{i},\mathbf{f}\_{j})+  \alpha_A \sum\_{i,j,k,l} T\_{i,j} T\_{k,l}\ell_A(\hat{A\}\_{i,k},A\_{j,l}) \newline &+ \rho \varphi(\mathbf{t}\_{1}, \hat{h} ) + \rho  \varphi(\mathbf{t}\_{2}, h )
> \end{align}
>
> Where $\mathbf{t}\_{1}$ and $\mathbf{t}\_{2}$ are the marginals of $\mathbf{T}$ and $\varphi$ is the divergence that controls the soft marginal constraints, for instance L2. Following [2], the solver solves a sequences of linear UOT problems for which a majoration-minimization (mm) algorithm is provided in [3].
>
> This UOT variant of PMFGW looks sound, but when we tried it we ran into the following issues:
>
> 1. The FUGW solver is extremely slow, about 10x slower than that of PMFGW. We tried to implement a GPU batched version but speedup was limited due to the shared convergence criterion.
> 2. Tuning the hyperparameters is very hard, the soft marginal constraint $\rho$ in particular is very unstable. We provide a simple illustration below for $\alpha_A = 0$ (that is we are only trying to learn the nodes).
>
>  Denoting
>
> $$\mathbf{T}^* = \text{argmin} \quad \alpha_f  \sum\_{i,j}T\_{i,j}\ell\_F(\hat{\mathbf{f}}\_{i},\mathbf{f}\_{j})+ \rho  \varphi(\mathbf{t}\_{1}, \hat{h} ) + \rho \varphi(\mathbf{t}\_{2}, h )   $$
>
> we know from equation (7) of [3] that $T^*_{i,j} = 0$ whenever $\alpha\_f \ell\_F(\hat{\mathbf{f}}\_{i},\mathbf{f}\_{j}) > \rho  (\hat{h}\_i + h\_j)$. In particular:
>
> $$\alpha\_f \ell\_F(\hat{\mathbf{f}}\_{i},\mathbf{f}\_{j}) \geq 2\rho \quad \implies \quad T^*\_{i,j} = 0$$
>
> This means that if $\rho$ is set too low it can happen that the optimal transport is $\mathbf{T}^* = 0$. But in that case the loss backpropagated to the network is
>
> $$\mathcal{L}(\hat{y},y) = \rho \varphi(\mathbf{0},\hat{h}) + \rho \varphi(\mathbf{0},h) = \rho \varphi(\mathbf{0},\hat{h}) + \text{cst}$$
>
> And the SGD will push the neural network toward a trivial prediction $\hat{h} = 0$.
>
> Because of the wide variability you mention we were never able to overcome such instabilities despite searching over a grid of parameters.
>
> [1] Thual, A., Tran, Q. H., Zemskova, T., Courty, N., Flamary, R., Dehaene, S., & Thirion, B. (2022). Aligning individual brains with fused unbalanced Gromov Wasserstein. Advances in neural information processing systems, 35, 21792-21804.
>
> [2] Séjourné, T., Vialard, F. X., & Peyré, G. (2021). The unbalanced gromov wasserstein distance: Conic formulation and relaxation. Advances in Neural Information Processing Systems, 34, 8766-8779.
>
> [3] Chapel, L., Flamary, R., Wu, H., Févotte, C., & Gasso, G. (2021). Unbalanced optimal transport through non-negative penalized linear regression. Advances in Neural Information Processing Systems, 34, 23270-23282.

---

> > ### Comment · Reviewer_bW5k · 2024-08-10
> > **Reply to the rebuttal**
> >
> > I appreciate the responses, especially the detailed answer regarding UOT as a loss. I will maintain the proposed score.

---

### Official Review · Reviewer_9w7t · 2024-07-18

**Soundness:** 3
**Presentation:** 3
**Contribution:** 3
**Rating:** 7
**Confidence:** 3

**Summary:**

This paper presents Any2Graph, a generic framework for end-to-end supervised graph prediction (SGP) with an optimal transport loss. The framework handles various input modalities and output graphs of arbitrary size and node ordering. The novel Partially-Masked Fused Gromov-Wasserstein loss is differentiable and permutation invariant, making it suitable for SGP. Numerical experiments showcase the versatility and superior performance of Any2Graph on a synthetic dataset and several real-world tasks such as map construction and molecule prediction. The paper addresses a practical challenge in deep learning models and offers a promising solution.

**Strengths:**

Innovative Methodology: The paper introduces a novel approach to supervised graph prediction (SGP) that is end-to-end, versatile, and achieves state-of-the-art results. The proposed framework, Any2Graph, leverages a new asymmetric Partially-Masked Fused Gromov-Wasserstein loss that is differentiable and node permutation invariant.
Practical Impact: The method is demonstrated on a wide range of real-world tasks including map construction from satellite images and molecule prediction from fingerprints. The results showcase the effectiveness of the proposed approach, offering a promising solution to practical challenges in deep learning models.

**Weaknesses:**

1. This paper addresses the SGP problem, characterized by size agnostic, node order insensitivity, and a vast search space. As the number of nodes increases, the computational load may significantly increase. However, the discussion of this issue in the paper is relatively limited.
2. Evaluation and Ablation Studies: The evaluation results are impressive but lack sufficient ablation studies. Additional analysis on the impact of different components (e.g., loss function) would enhance our understanding of the proposed method’s strengths and weaknesses.

**Questions:**

1. Can an analysis of computational complexity be provided? Can a computationally feasible solution be proposed for graph generation problems with a large number of nodes (e.g., graphs with more than 100 nodes, which are commonly encountered in various data sets)?
2. Have you considered performing ablation studies or comparisons against popular baselines?
3. Can you discuss potential limitations or failure cases of your method? How might these impact real-world applications? Are there scenarios where traditional methods may outperform Any2Graph?

**Limitations:**

As Discussed In Sec. Weakness & Sec. Limitations

---

> ### Author Rebuttal · Authors · 2024-08-06
>
> Thank you for taking the time to review our paper. We adress your many interesting questions below:
>
> > 1. Can an analysis of computational complexity be provided? Can a computationally feasible solution be proposed for graph generation problems with a large number of nodes (e.g., graphs with more than 100 nodes, which are commonly encountered in various data sets)?
>
> We fully agree that the original version of the paper was missing a detailed computational complexity analysis that we have now added. Thank you for this important remark. In short: the cost of the transformer scales with $M^2$ and the cost of PMFGW scales with $kM^3$ where $k$ is the number of iteration required for the solver. We provide a novel plot for empirical estimation of $k$ in the pdf. Thus, this version of Any2Graph, like Relationformer, cannot scale to graphs with hundreds of nodes. Our opinion is that the supervised prediction of small graphs (few tens of nodes) is already a very challenging topics with many real world application. The goal of this first work is not to scale beyond this order of magnitude but to introduce a novel framework. Yet, we agree that this is a natural question and this motivated us to augment the conclusion with a detailed plan to scale Any2Graph in future works (using approximate attention [1], heuristics [2] and entropic regularization [3]).
>
>
> > 2. Have you considered performing ablation studies or comparisons against popular baselines?
>
> This is a natural question but we don't think that is possible to remove any part of Any2Graph without fully breaking the pipeline. For instance if we remove the $l_h$ terms in PMFGW, the model will simply not learn to predict which nodes are activated and the accuracy will drop to 0. Yet, we would like to point out that the comparison with Relationformer can be seen as a form of ablation study, since we use the exact same architecture except for the loss. We also provide the results with and without feature diffusion which we see as a form of ablation study even if we don't formulate it that way due to space constraints.
>
> > 3. Can you discuss potential limitations or failure cases of your method? How might these impact real-world applications?
>
> This is very important as well, thank you. We think that there are two types of failure cases. On the one hand, it can happen that the training dynamic fails to converge. We discuss this in detail in appendix F.1 along with simple methods to prevent this bad behaviour. On the other hand there is the case where $M$ is too large which we already discussed above.
>
> > 4. Are there scenarios where traditional methods may outperform Any2Graph?
>
> Split from the previous question for readability. To the best of our knowledge, the surrogate regression methods are the 'tradionnal approach' for tackling SGP in a general fashion. Yet they operate on a different data regime than Any2Graph as they are able to deal with scarse data but come with a high decoding cost. For instance, Any2Graph is ill-suited to deal with the metabolite prediction task [4] that benefits from a known candidate graph set.
>
> [1] Fournier, Q., Caron, G. M., & Aloise, D. (2023). A practical survey on faster and lighter transformers. ACM Computing Surveys, 55(14s), 1-40.
>
> [2] D. B. Blumenthal et al. “Comparing heuristics for graph edit distance computation”. The VLDB journal
>
> [3] Altschuler, J., Niles-Weed, J., & Rigollet, P. (2017). Near-linear time approximation algorithms for optimal transport via Sinkhorn iteration. Advances in neural information processing systems, 30.
>
> [4] Brogat-Motte & all. (2022). Vector-valued least-squares regression under output regularity assumptions. Journal of Machine Learning Research

---

> > ### Comment · Reviewer_9w7t · 2024-08-08
> > **Thanks!**
> >
> > I appreciate the response and all generally makes sense. I'll maintain my score.

---

### Official Review · Reviewer_YNCo · 2024-07-18

**Soundness:** 3
**Presentation:** 3
**Contribution:** 2
**Rating:** 7
**Confidence:** 4

**Summary:**

This work proposes Any2Graph, an end-to-end deep learning framework for Supervised Graph Prediction (SGP) leveraging a novel OT loss called PMFGW. The model consistently achieves state-of-the-art performances across multiple graph prediction tasks and input modalities.

**Strengths:**

- The author combines Partial Matching and OT methods, using permutation and padding operator to consistently predict and compare graphs of arbitrary sizes.
- The author uses a variety of datasets to illustrate that Any2Graph has consistent and strong performance on different modal inputs, including noisy images, real world satellite images and texts (molecular fingerprints).

**Weaknesses:**

- There are some inconsistent notations in the paper. For example, the loss function for node features is denoted as $l_F$ in Eq. (1), while the author uses $l_f$ in the next sentence.
- The author emphasizes the importance of the maximum node size $M$ that strongly influences efficiency and expressiveness of the model. However, the analysis in Section 5.3 shows $M=16$ is sufficient to tackle the problem, indicating that the datasets may be relatively simple and cannot fully reflect the prediction ability on more complex graphs with higher orders of magnitude of nodes.

**Questions:**

- The author uses PMFGW as the evaluation metric at the graph level, which is also the training objective of the model. The comparison may be not fair in that case since the loss functions $l_h, l_f, l_A$ and weights $\alpha$ can be arbitrarily chosen. Please correct me if I am wrong.
- In Section 4, the author mentions that threshold 1/2 is chosen. I wonder whether and how the threshold influences the model's capability. Please show me some experimental results on one or more datasets.
- The datasets may be relatively simple since $M=16$ is sufficient to tackle the problem. Can you give some results on more complecated datasets (\emph{i.e.}, with higher orders of magnitude of nodes)?

**Limitations:**

- As mentioned in the paper, the main limitation is its scalability to graphs of larger size.

---

> ### Author Rebuttal · Authors · 2024-08-06
>
> We are greatful for the in-depth review of our paper. We also thank you for having pointed out typos. They will be corrected in the final version of the paper. We answer your questions below:
>
> > The author uses PMFGW as the evaluation metric at the graph level, which is also the training objective of the model. The comparison may be not fair in that case since the loss functions $l_h, l_F, l_A$ and weights can be arbitrarily chosen. Please correct me if I am wrong.
>
> You are absolutetly correct. This is why we never use PMFGW as an objective metric in the experiments. That being said, we were very careful to report meaningful values in table 1 by fixing $l_h, l_F, l_A$ and weights for each tasks, the values used are that reported in appendix E.2. We will make this clear in the final version of the paper.
>
> > In Section 4, the author mentions that threshold 1/2 is chosen. I wonder whether and how the threshold influences the model's capability. Please show me some experimental results on one or more datasets.
>
> This is an interesting point. We provide a few plots with different thresholds in the global response pdf. As you can see the model is very robust to the choice of the threshold as it is quite confident in its prediction.
>
> > The datasets may be relatively simple since $M=16$ is sufficient to tackle the problem. Can you give some results on more complecated datasets (i.e., with higher orders of magnitude of nodes)?
>
> Our opinion is that the supervised prediction of small graphs (few tens of nodes) is already a very challenging problem with many realy world applications. The goal of this first work is not to scale beyond this order of magnitude. That being said, your question motivated us to train Any2Graph on a dataset with graphs of size up to size $50$ (figure 2). Still, we understand that scaling to larger graph is a natural question. Thus we have enriched the conclusion of the paper with more ideas for scaling Any2Graph in a future work (using approximate attention [1], heuristics [2] and entropic regularization [3]).
>
> [1] Fournier, Q., Caron, G. M., & Aloise, D. (2023). A practical survey on faster and lighter transformers. ACM Computing Surveys, 55(14s), 1-40.
>
> [2] D. B. Blumenthal et al. “Comparing heuristics for graph edit distance computation”. The VLDB journal
>
> [3] Altschuler, J., Niles-Weed, J., & Rigollet, P. (2017). Near-linear time approximation algorithms for optimal transport via Sinkhorn iteration. Advances in neural information processing systems, 30.

---

> > ### Comment · Reviewer_YNCo · 2024-08-08
> >
> > I'm quite appreciated for detailed explanations and additional experiments provided in the attachment. Results have demonstrated that the method is suitable for more complicated graphs with $M=50$ (though it could be better to show concrete values of the metric in the final version) and is robust to different threasholds as well. After reading the comments of other reviewers and the author's rebuttal, I'd like to change the rating to 7.
> >
> > Wish you all the best.

---

### Official Review · Reviewer_CHfS · 2024-07-18

**Soundness:** 3
**Presentation:** 3
**Contribution:** 2
**Rating:** 6
**Confidence:** 3

**Summary:**

This work aims to design an end-to-end pipeline for structured graph prediction (SGP). The proposed loss, PMFGW, is a extension of Fused Gromov Wasserstein to generate graphs with bounded arbitrary sizes, along with a standard pipeline carefully modified from Relationformer. It is empirically verified on public tasks and a novel synthetic dataset, Coloring, with outstanding performance.

**Strengths:**

1. The solution is straightforward with good empirical performance: the extension is clear as penalizing loss for misalignment of padding dependent of graph sizes, and the empirical performance is outstanding against the related work, Relationformer.
2. The presentation is well written: for each contribution and novel design, the details are closely accompanied with reasons of design and necessary discussion about related works, which makes it easy to position this work in the literature.

**Weaknesses:**

One may argue that a clearly successful paper could have more novelty, than proposing a loss with an additional term, the significance of which I am not sure about. The positive side is that the simple design has good empirical performance, and perhaps could inspire other researchers to simply add such a term to solve graph tasks with different sizes.

Hence, I would like to provide a score of weak accept for now, and see how the other reviewers think about this potential argument.

**Questions:**

1. Could you please reveal more why the reported results about Relationformer in Table 1 are largely different from their original ones? I suppose the reason may be as written around line 278 ''use the same architecture for both approaches...'', but I still want to make sure the difference is well understood. Is there any other issue here, like data split?

**Limitations:**

Yes.

---

> ### Author Rebuttal · Authors · 2024-08-06
>
> Thank you for reviewing our paper! Note that we moved the following discussion in the global response as it might interest the other reviewers:
>
> > One may argue that a clearly successful paper could have more novelty, than proposing a loss with an additional term, the significance of which I am not sure about. The positive side is that the simple design has good empirical performance, and perhaps could inspire other researchers to simply add such a term to solve graph tasks with different sizes.
>
> We would like to further explain our point of view here. We agree that PMFGW is a relatively straightforward extension of an existing OT problem but using it for SGP is novel and we hope that the important gain in performance of such a natural framework will help to spark more interest into SGP. This is also the reason why we provide tools for benchmarking future methods (synthetic datasets and set of metrics). Finally, note that we explored significantly more complex approaches with weaker results (see the UOT discussion with reviewer **bW5k**).
>
> > Could you please reveal more why the reported results about Relationformer in Table 1 are largely different from their original ones? I suppose the reason may be as written around line 278 'use the same architecture for both approaches...', but I still want to make sure the difference is well understood. Is there any other issue here, like data split
>
> This is an important point indeed. You are correct, the main difference comes from the architecture as we use a simple Resnet18 while relationformer leverage a Resnet50 and a complex MultiLevel Attention scheme. This is very specific to images inputs and we did not use it as we wanted to show the generality of our approach to different data modalities. The data splits however are exactly the same (those of the original Toulouse and USCities papers). Finally, the definition of some metrics differ between the two papers. The graph level metrics reported in relationformer rely on the TOPO and SMD score which are heavily specific to Sat2Graph tasks. We believe that the Node level and Edge level metrics are similar to ours but no exact definition are given in the transformer paper. Anyway, the final table we provide is a fair comparison relying on the same backbone and a set of well defined task-agnostic metrics.

---

> > ### Comment · Reviewer_CHfS · 2024-08-12
> >
> > Thanks for your response. After reading the response, I will maintain my score for ''Technically solid, moderate-to-high impact paper''.

---

### Author Rebuttal · Authors · 2024-08-06

First we would like to thank the reviewers for their mostly positive reviews with constructive questions.

The majority of the comments are positive with many reveiwers finding the paper well written (**CHfS**,**bW5k**,**ct8H**), well positioned in the literature (**CHfS**) and with a nice potential impact in SGP and applications (**ct8H**,**bW5k**). Reviewers also noted that the numerical experiments are solid (**ct8H**, **bW5k**) and demonstrate very good empirical performance for a wide range of modalities (**CHfS**,**YNCo**,**9w7t**,**YNCo**), all this with very low computational inference cost (**bW5k**). The novel proposed loss PMFGW remain simple (**CHfS**) but also relies a sound theoretical analysis (**bW5k**).

On a slightly more reserved note, **CHfS** was wondering about the limited novelty of the PMFGW loss but we believe that those small changes are the ones that made the large gain in performance possible which provides an important contribution to the community. Reviewers also asked questions about the scalability of the method (**YNCo**, **9w7t**, **ct8H**). We acknowledged that the computationnal complexity analysis was not detailed enough and thus provided new experiments to fill this gap. Any2Graph is suited to the supervised prediction of graphs with few tens of nodes (we provide a novel exemple with 50 nodes) and reviewers agree that this is enough for many practical applications (**ct8H**, **bW5k**). Yet we understand that scaling to larger graph is a natural question. For that reason, we enriched the conclusion of the paper by drawing additional ideas for scaling Any2Graph in a future work.

---

### Decision · Program_Chairs · 2024-09-25

**Decision:**

Accept (spotlight)

**Comment:**

This paper presents Any2Graph, a versatile end-to-end deep learning framework designed for Supervised Graph Prediction (SGP). The core innovation of this framework is the introduction of a novel loss function, Partially-Masked Fused Gromov-Wasserstein (PMFGW), which extends the Fused Gromov-Wasserstein loss to accommodate graph prediction with arbitrary sizes and node orderings.  Along with a standard pipeline carefully modified from Relationformer, the framework demonstrates consistent and state-of-the-art performance across a diverse range of datasets, including real-world tasks like map construction from satellite images and molecule prediction from fingerprints.

The reviewers praise the paper regarding its novelty, theorectical soundness, strong empirical performance, and clearity. The PMFGW loss is highlighted as a significant contribution, providing a differentiable, permutation-invariant, and size-agnostic approach to graph prediction, which is a meaningful extension of existing methods. Some minor issues were raised regarding the presentation, experimental evaluations, and complexity analyses. However, all reviewers were satisfied with the authors' explanations and provided positive support for the paper. The paper is recommended for acceptance.